# Multifaceted Evaluation of Inhibitors of Anti-Apoptotic Proteins in Head and Neck Cancer: Insights from In Vitro, In Vivo, and Clinical Studies (Review)

**DOI:** 10.3390/ph17101308

**Published:** 2024-09-30

**Authors:** Kamil Krzykawski, Robert Kubina, Dominika Wendlocha, Robert Sarna, Aleksandra Mielczarek-Palacz

**Affiliations:** 1Department of Immunology and Serology, Faculty of Pharmaceutical Sciences in Sosnowiec, Medical University of Silesia in Katowice, 41-200 Sosnowiec, Poland; dwendlocha@sum.edu.pl (D.W.); apalacz@sum.edu.pl (A.M.-P.); 2Silesia LabMed: Centre for Research and Implementation, Medical University of Silesia in Katowice, 41-752 Katowice, Poland; rkubina@sum.edu.pl (R.K.); robert.sarna@sum.edu.pl (R.S.); 3Department of Pathology, Faculty of Pharmaceutical Sciences in Sosnowiec, Medical University of Silesia in Katowice, 41-200 Sosnowiec, Poland

**Keywords:** apoptosis, IAP, HNSCC, smac mimetics, cancer, head and neck

## Abstract

This paper presents a multifaceted assessment of inhibitors of anti-apoptotic proteins (IAPs) in the context of head and neck squamous cell carcinoma (HNSCC). The article discusses the results of in vitro, in vivo, and clinical studies, highlighting the significance of IAPs in the resistance of cancer cells to apoptosis, which is a key factor hindering effective treatment. The main apoptosis pathways, including the intrinsic and extrinsic pathways, and the role of IAPs in their regulation, are presented. The study’s findings suggest that targeting IAPs with novel therapies may offer clinical benefits in the treatment of advanced HNSCC, especially in cases resistant to conventional treatment methods. These conclusions underscore the need for further research to develop more effective and safer therapeutic strategies.

## 1. Head and Neck Cancer

Head and neck squamous cell carcinoma (HNSCC) is a heterogeneous group of cancers that includes lesions located in the oral cavity, oral part of the pharynx, lower pharynx, larynx, nasopharyngeal cavity, and sinuses [1,2]. According to the latest GLOBOCAN data, 947,211 people would develop HNSCC in 2022, and 482,428 would die. These alarming statistics clearly indicate that HNSCC is a serious global health problem, underscoring the urgent need to develop more effective methods of treating and preventing the disease [3].

Currently, smoking and using other tobacco products, as well as chronic alcohol consumption, especially in combination with smoking, are considered major risk factors worldwide (Figure 1) [1,4,5]. In recent years, electronic nicotine delivery systems, including e-cigarettes and vaporizers, have experienced a significant rise in popularity [6,7]. In vitro studies indicate that their use induces oxidative stress, leads to DNA damage such as double-strand breaks, and triggers the processes of necrosis and apoptosis. Additionally, it has been observed that they may promote more aggressive phenotypes in cancer cells [8,9]. However, due to the relatively short time since their introduction to the global market (2006), assessing their long-term health effects is currently not possible, and the available evidence regarding their potential oncogenic effects remains limited [6,8,10]. The other major important cause of cancerous lesions in the head and neck region is high-risk human papillomavirus (HPV) infection [1,4,11]. Of the various types of HPV, types 16 (the vast majority of cases) and 18 are responsible for more than 90% of HPV(+) HNSCCs, whose locations are mainly the oral cavities and throat and whose incidence varies significantly by geographic location [1,4,12]. In high-income countries such as the US and Western European countries, the incidence of HPV(+) HNSCC is high, accounting for about 60–70% of HNSCCs, while in low- and middle-income countries, the figure is likely to be less than 10% [12,13]. The number of HPV(−) HNSCCs associated with chronic exposure to tobacco smoke and alcohol, especially in the elderly, is showing a slow downward trend associated with a gradual reduction in tobacco use. Unfortunately, at the same time, the number of new cases of HNSCC HPV(+) in younger people is increasing rapidly [11]. Not only does HPV constitute a significant cause of HNSCC, but other viruses, such as Epstein–Barr virus (EBV), also play a crucial role in oncogenesis [1,14]. EBV is the most oncogenic among the herpesviruses and has been classified by the International Agency for Research on Cancer as a Group I carcinogen [14]. EBV has been detected in various malignancies, including nasopharyngeal carcinoma, Hodgkin’s lymphoma, Burkitt’s lymphoma, gastric cancer, and laryngeal cancer. The virus is transmitted through saliva and infects over 90% of the adult population globally, which, combined with its ability to induce epigenetic changes, poses a significant health threat [14,15]. Herpesviruses such as herpes simplex virus type 1 (HSV-1), which causes oral herpes, and cytomegalovirus (CMV) also exhibit potential oncogenic properties. Although their direct role in cancer development is not definitively established, it is suspected that they may contribute to carcinogenesis through chronic inflammation, interaction with other carcinogenic factors, and the modulation of the immune response [14]. Other risk factors for HNSCC include genetic predisposition, a diet low in fruits and vegetables and high in animal fats, air pollution, pharyngeal and laryngeal reflux, Fanconi anemia (FA), areca nuts, previous radiation therapy, and certain harmful factors found in the work environment [1,4,16].

HNSCCs include clinically, molecularly, and histologically differentiated cancers that are treated with similar methods, often with limited success. Standard therapeutic management for both HPV(+) and HPV(−) HNSCC patients includes surgical resection, chemotherapy, and radiation therapy. Unfortunately, all these methods carry a high risk of side effects, often lack efficacy, and lead to a serious reduction in patients’ quality of life [5]. In patients with HNSCC at an early, nonadvanced stage unrelated to HPV infection, radiotherapy or surgery can achieve a five-year survival rate of 70–90% [4,11,12]. The situation differs for advanced-stage patients with primary disease at the T3–T4 level according to the TNM classification or with involved cervical lymph nodes, accounting for about 60% of diagnosed patients. The five-year survival rate then ranges from 49% to 25%, depending on the location of the primary disease [11,12]. HNSCCs caused by HPV infection, despite the increased appearance of metastases in the cervical lymph nodes, have a more favorable prognosis but frequent recurrences, which can occur up to five years after receiving the diagnosis, which affects 10–25% of patients [12,13,17,18].

The lack of characteristic symptoms and the varied clinical presentation of HNSCC poses significant challenges in diagnosis [19]. These tumors are most often detected incidentally by a primary care physician or dentist, and only at a later stage is the patient referred to a specialist [4,19]. An otolaryngologist or head and neck surgeon evaluates the oral cavity, oropharynx, larynx, and nasopharynx, and if necessary, refers the patient for imaging of the head, neck, and chest, as well as histopathological analysis of the tissue sample [4,20]. The currently used diagnostic methods are time-consuming, invasive, and difficult to interpret [20,21]. Consequently, there is a need to develop new, accurate, and rapid screening methods for HNSCC [19,22]. The most significant and approved biomarkers for HNSCC include HPV status, interleukin-8, and p16 protein [21]. Current research is focused on the development and validation of additional biomarkers found in blood, saliva, and exhaled breath [19,22,23]. Advances in nanotechnology allow for the detection of prognostic markers at low concentrations and improve the capabilities of currently used imaging techniques [21]. Quantum dots (QDs) represent a particularly promising avenue in cancer diagnostics due to their unique optical properties. They are characterized by broad absorption spectra and narrow emission spectra, which can be precisely tuned by modifying their size and composition. QDs also exhibit high resistance to photobleaching, a significant Stokes shift, and high fluorescence efficiency [24,25,26]. Due to their small size, quantum dots can penetrate cancer cells, enabling precise tumor tracking and imaging. Key clinical applications of QDs include the detection of micrometastases and sentinel lymph node mapping [24,26]. Research is also ongoing regarding the use of artificial intelligence (AI) and machine learning (ML) to create algorithms for the analysis of imaging and molecular data. The application of AI/ML may, in the future, improve the effectiveness of imaging and molecular data analysis, as well as enable the automation of this process [20,27]. Due to unsatisfactory treatment results and the significant toxicity of therapies used in patients with advanced HNSCC and metastatic or recurrent disease, the development of therapeutic options other than palliative chemotherapy has become a priority [28].

In 2011, the Food and Drug Administration (FDA) approved cetuximab, a monoclonal antibody against the epidermal growth factor receptor (EGFR), as a treatment for recurrent and metastatic HNSCC (R/M HNSCC) [2,28,29]. Unfortunately, despite the very common overexpression of GFR in HNSCC cells, cetuximab and other EGFR-targeted therapies have failed to demonstrate satisfactory efficacy [5]. Five years later, in 2016, the FDA approved nivolumab and pembrolizumab, which are immune checkpoint inhibitors of programmed cell death (PD-1), for the treatment of patients with R/M HNSCC refractory to treatment with platinum compounds [2,28]. Subsequently, the FDA approved pembrolizumab in combination with platinum and 5-fluorouracil in the same population as a monotherapy for populations of patients with HNSCC shown to have an increased expression of programmed death receptor ligand 1 (PD-L1) [30]. Despite the favorable toxicity profile of immunotherapeutic agents compared to standard chemotherapy, only a small group of patients can count on sustained remission or improved survival from treatment with immune checkpoint inhibitors [4,11]. An estimated 90% of patients with R/M HNSCC will not respond to treatment [11]. The limitations of therapies targeting programmed death receptor 1 (PD-1) and EGFR, as well as the strong adverse effects of standard treatments, necessitate the development of new, safer treatments for R/M HNSCC [4,11,31]. One of the potential directions is the use of photodynamic therapy with QDs. These nanomaterials are characterized by high selectivity, allowing for the precise targeting of cancer cells while minimizing the risk of damage to healthy tissues. Through the mechanism of Förster resonance energy transfer, QDs efficiently generate reactive oxygen species, leading to the destruction of cancer cells. Additionally, their advanced structure enhances the stability and bioavailability of photosensitizers, prolonging their action and increasing treatment efficacy. Furthermore, quantum dots enable simultaneous diagnosis and therapy, allowing for the real-time monitoring of the treatment process. These features make the combination of PDT with QDs an innovative, more effective, and safer method of cancer treatment, while also minimizing side effects [24,32,33].The goal of anticancer therapies is to induce apoptosis, but oncogenic changes in cells reduce their clinical effectiveness. Targeted and personalized therapies are therefore needed. The molecular heterogeneity of HNSCC makes the clinical response difficult to predict, which poses a significant challenge for oncologists and appears to be a key factor affecting treatment outcomes. Much of this clinical heterogeneity is likely due to changes in pathways leading to cell death [34].

## 2. Apoptosis

The primary process for eliminating damaged and unwanted cells and maintaining cell numbers in balance is apoptosis [35,36]. The abnormal regulation of this process underlies a wide variety of diseases and represents a promising target for new anticancer therapies. The activation of apoptosis occurs via two major, well-characterized pathways: the intrinsic or mitochondrial pathway and the extrinsic pathway (Figure 2) [36,37].

The intrinsic apoptosis pathway depends on factors released from mitochondria and can be activated by both positive and negative pathways [36]. Activation by the positive pathway occurs due to factors such as DNA damage, hypoxia, chemotherapeutic drugs, radiation, viruses, reactive oxygen species, and toxins [35,36,37]. The mitochondrial pathway can also be activated through a negative pathway by the absence of signals that promote survival in the form of cytokines, growth factors, and hormones [36].

During the activation of the intrinsic apoptotic pathway, the outer mitochondrial membrane is permeabilized, resulting in the release of cytochrome c into the cytosol [35,36,37,38]. This is favored by proapoptotic proteins belonging to the BCL-2 family, such as BAK, BAX, and PUMA. These proteins can be divided into sensitizers (e.g., BIK, BAD, and NOXA) and activators (e.g., PUMA, BID, and BIM). Activators bind to BAK and BAX proteins, which are the main effectors of apoptosis and cause their oligomerization, leading to the release of cytochrome c. Sensibilizers promote the apoptosis process by controlling the monomeric localization of BAK and BAX and inhibiting antiapoptotic factors [38]. The BCL-2 family of proteins also includes antiapoptotic proteins such as BCL-2 or BCL-XL that have two BCL-2 homology domains, which inhibit activators and enable the maintenance of a balance between the resistance and sensitivity of cells to apoptosis. Cytochrome c induces apoptosis by binding the cytosol to an adapter molecule, apoptotic protease activating factor 1 (APAF1), which causes its conformational change—because of this, the nucleotide binding domains are exposed, and oligomerization occurs. Due to these changes, it becomes possible to bind deoxyadenosine triphosphate (dATP), which leads to another conformational change involving the exposure of the caspase recruitment domain (CARD) and the oligomerization domain, resulting in the formation of a complex composed of several APAF1 units called the apoptosome. The apoptosome exposes CARD domains in its active center, allowing for the further recruitment and activation of procaspase-9 [36]. The resulting caspase-9 then cleaves and activates effector caspases-3 and -7, thereby inducing apoptosis [35,37].

The extrinsic apoptosis pathway is initiated by binding death ligands, which include tumor necrosis factor α (TNFα), the TNF-related apoptosis-inducing ligand (TRAIL/Apo2L), and the Fas ligand (FasL/Apo-1) [37,38]. These ligands bind to their corresponding death domain (DD)-containing receptors TNFR1, DR4/DR5, and CD95/FasR, respectively, located on the cell surface [37]. Upon the activation of the aforementioned receptors, the oligomerization of the receptors occurs, followed by the recruitment of adapter proteins, such as FADD and TRADD, and the formation of a death-inducing signaling complex (DISC) in the cell membrane [35,36,38]. The DISC recruits procaspase-8 and -10, which are mutually cleaved and activated [36,37,38]. Activated caspase-8 and -10 further activate effector caspases (i.e., caspase-3, -6 and -7), thus directly initiating apoptosis [35,36,37,38].

The mitochondrial and extrinsic pathways are closely linked. Activated caspase-8 cleaves a member of the BCL-2 family, BID, and leads to the formation of truncated BID (tBID), which then moves to the mitochondrial membrane, and through binding to BAK and BAX, it promotes membrane permeabilization, resulting in the release of cytochrome c into the cytoplasm and the activation of the intrinsic apoptosis pathway [37,38]. Common to both apoptosis pathways is the activation of effector caspases, which can also activate caspase-8, and a positive feedback loop for apoptosis is then formed [37]. During the permeabilization of the outer mitochondrial membrane, in addition to cytochrome c, a second mitochondrial caspase activator (Smac/Diablo) and a high-temperature A2 protein (HtrA2/Ommi)—natural antagonists of the apoptosis inhibitor family of proteins—are also released into the cytoplasm [36,37,39].

## 3. Inhibitors of Apoptosis Proteins

Proteins in the inhibitor of apoptosis (IAP) family are suspected to be potential targets for new targeted therapies and one of the causes of cancer cell resistance [40,41]. The IAP family includes antiapoptotic regulators that block cell death in response to various stimuli through interactions with inducers and effectors of apoptosis. These proteins are overexpressed in most malignancies, playing a key role in tumor maintenance by inhibiting cell death and multiple signaling pathways [42]. IAPs are found in many organisms and were originally discovered in baculovirus in studies aimed at determining which viral proteins prevent infected cell death [43]. In viruses, this effect is achieved through direct interaction with caspases, whereas in humans, only one IAP representative shows a strong ability to directly bind and inhibit caspases. The other representatives achieve their antiapoptotic effect through the regulation of cell signaling, related to sequence differences in the positions exposed on the surface, which alter the binding properties [40,41,43,44]. We can identify IAPs based on the presence of at least one BIR (baculovirus IAP repeat) domain containing three cysteine residues and one histidine residue coordinating a Zn^2+^ ion of 70–80 amino acids in length (Figure 3), involved in protein–protein interactions [40,41,43,44,45]. In humans, eight members of the IAP family have been described: X chromosome-associated IAP (XIAP), neuronal apoptosis inhibitory protein (NAIP), cellular IAP1 (c-IAP1), cellular IAP2 (cIAP2), IAP2-like protein (ILP2), melanoma IAP (ML-IAP), survivin, and apollon, of which XIAP and cIAP1/2 are considered the most crucial [45,46].

XIAP is the only IAP molecule showing a strong affinity for the direct binding and inhibition of caspases [41]. The activity of effector caspases, specifically caspase-3 and -7 (Figure 4), is inhibited by binding the BIR1-BIR2 linker region of XIAP to their active sites in a reverse orientation, thus preventing the cleavage of the linker. Further, the connection between the linker and caspase is stabilized by binding the BIR2 domain to the exposed IAP-binding motif. This interaction means that access to substrates remains restricted, and caspases cannot perform their functions. The BIR3 domain of XIAP binds to the IAP-binding motif of caspase-9, located just above the active site, the exposure of which occurs during the proteolytic transformation of pro-caspase-9 and results in restricted access to substrates [40,44,48,49]. Active caspases are dimers, so the action of the BIR3 domain leading to caspase-9 monomerization reverses the activation mechanism and consequently leads to its inhibition [40,44,50].

The BIR1 domain is involved in NF-κB pathway activation and XIAP conformational changes [46]. In addition to BIR, most IAPs proteins also have other functional domains [50]. A structural element found in XIAP, ML-IAP, and cIAP1/2 is the domain of an interesting new gene, RING, which is characterized by E3 ubiquitin ligase activity. RING enables the auto- or trans-ubiquitination of bound molecules, which can consequently lead to proteasomal degradation or alter the signaling properties of bound proteins [45,51]. In the structure of XIAP and cIAP1/2 proteins, we can also distinguish a conserved ubiquitin-associated (UBA) domain, which allows host proteins to participate in ubiquitin-dependent signaling processes [52].

The cIAP1 and cIAP2 proteins also have a conserved CARD in their structure, which serves as a surface for interacting proteins, although its role in IAPs is not yet fully understood [53]. Due to the similarity in structure and organization of the cIAP1 and cIAP2 protein domains relative to XIAP, it was originally thought that cellular IAPs were also inhibitors of caspases. However, later studies have shown that cIAPs play their role in inhibiting apoptosis in a slightly different way [44].

CIAP1 and cIAP2 are potent inhibitors of the extrinsic apoptosis pathway [45]. They also regulate classical and alternative signaling of the nuclear factor kappa-light-chain-enhancer of activated B cells (NF-κB) [54]. NF-κB refers to a family of transcription factors that regulate many biological processes, including cell growth and survival and inflammation [55]. The common feature of the five members of the NF-κB family (i.e., RelA (p65), RelB, c-Rel, p100/p52, and p105/p50) is the presence of a REL homology domain at the amino end. NF-κB activation can occur via two distinct signaling pathways (i.e., the canonical and noncanonical NF-κB pathways) [55,56]. CIAPs negatively regulate NF-κB inducible kinase (NIK) in the steady state and prevent its accumulation (Figure 5) [54,55]. The inhibition of the noncanonical NF-κB pathway is mediated by the cIAP-TRAF2-TRAF3 complex, which causes NIK ubiquitination and subsequent proteasomal degradation [56]. The abolition of cIAP1/2 function in the presence of IAP antagonists leads to stabilization and increased levels of NIK, followed by the phosphorylation of I kappa B kinase alpha (IKKα) [51,54]. IKKα subsequently phosphorylates p100 by partial proteasomal degradation to p52 [55]. The resulting RelB and p52 complex then translocates to the cell nucleus, where it participates in the regulation of gene transcription [55,56]. The canonical activation of NF-κB is promoted by TNFR1 and RIP1 ubiquitination [54,55,56]. RIP1 ubiquitination also prevents its association with the apoptotic FADD-caspase8 complex and prevents it from further binding to TNF superfamily receptors [54]. Ligand attachment to the TNFR1 receptor leads to the recruitment of a complex composed of cIAP1/2, TRADD, TRAF2, and RIP1. The ubiquitinated form of RIP1 is then recruited to the NFκB essential modulator protein, which leads to the formation of the TAK1-IKK complex [56]. TAK1 then activates IKKβ and consequently leads to the degradation of IκBα, allowing the p50-RelA complex to translocate into the cell nucleus and transcribe genes [54,55,56]. The inhibition of cIAP1/2 protein activity results in an auto/paracrine TNFα loop associated with the upregulation of NF-κB target genes [41]. Such a situation leads to the TNFR1-mediated activation of caspase-8 and consequently the entry of cells into the apoptotic pathway [51].

## 4. IAP Antagonists

The best known endogenous IAP antagonist is Smac [37]. At the N-terminus of Smac is a sequence composed of 55 amino acid residues, which undergoes proteolytic cleavage upon Smac release from the mitochondrion and results in the exposure of a key Ala-Val-Pro-Ile tetrapeptide motif (AVPI), serving to bind to cIAP1/2 and XIAP. IAP domains responsible for interactions with caspases (i.e., BIR2 and BIR3) can also interact with Smac [37,57]. Smac, upon its release from mitochondria, dimerizes and then binds to the BIR2 and BIR3 domains of XIAP, consequently preventing XIAP from effectively inhibiting caspases. Due to the presence of an AVPI-binding motif in the BIR domain of cIAP1 and 2, their inhibition is possible. Importantly, Smac promotes auto-ubiquitination and consequently the proteasomal degradation of cellular IAPs (i.e., cIAP1, and cIAP2) [37].

The identification of the structures of the endogenous Smac protein involved in Smac–IAP interaction enabled the design and development of so-called Smac mimetics (SMs) or small-molecule IAP antagonists [57,58]. SMs mimic the N-terminus of endogenous Smac, which has a four-amino-acid AVPI sequence. This element is crucial for Smac binding to the BIR2 and BIR3 domains of IAPs [57,59,60]. Most SMs developed to date can antagonize XIAP, cIAP1 and cIAP2 [59,61]. Importantly, different SMs may differ in the potency with which they inhibit specific IAPs, thus differing significantly in their antitumor effects [59]. Under physiological conditions, endogenous Smac forms a homodimer, while SMs can be divided into monovalent and divalent compounds [59,62]. Divalent Smac comprises two monomeric units joined by a chemical linker. These compounds have a higher affinity for IAPs, a greater potency to antagonize IAPs, and, consequently, stronger antitumor activity compared to monovalent compounds. However, they are characterized by lower bioavailability in the body [35,59]. SMs cause the activation of caspases-3, -7, and -9 by binding and limiting the function of XIAPs in competition for their respective binding sites. SMs also bind to cIAPs, and this interaction leads to stimulation of their E3 ligase, resulting in ubiquitination and degradation in proteasomes [58,59,60]. Depletion of cIAP1 and cIAP2 proteins leads to activation of the noncanonical NF-κB pathway [59,60]. Loss of cIAP results in stabilization and accumulation of NIK, which then phosphorylates and activates the IKK complex. Activated IKK is involved in the phosphorylation of the p100 protein, resulting in the formation of p52, which travels to the nucleus and stimulates the transcription of NF-κB target genes such as TNFα [58,59,60]. TNFα is then secreted into the extracellular space and binds in an auto/paracrine manner to TNF receptors [59,60]. Once TNFα binds to the receptor, the formation of a cell death complex made up of RIP1 protein, FADD, and caspase-8 is triggered, ultimately leading to the promotion of apoptosis [58,59,60].

In this review, we present the research to date on SMs in the context of head and neck cancer. We include the results of studies conducted in vitro, animal experiment results, and analyses from clinical trials.

## 5. In Vitro Studies

### 5.1. AZD5582

AZD5582 is a dimeric SM showing high binding strength to the BIR3 domains of XIAPs and cIAPs (Figure 6) [63]. In a study on HNSCC cells of the SCC25, Cal27, and FaDu lines, AZD5582 was applied at concentrations ranging from 0.93 to 15 µM, and its cytotoxic activity was shown to be dose-dependent. The efficacy of AZD5582 in combination in with radiotherapy at doses of 0, 2, 4, and 8 Gy was evaluated. The combination therapy produced diverse effects, producing a strong synergistic effect for the Cal27 cell line, an additive effect for FaDu cells, and an antagonistic effect for the SCC25 line. The combination therapy also induced long-term effects (i.e., a reduction in the ability of cells to form colonies). The effect of AZD5582 in all HNSCC cell lines tested resulted in an increase in the fraction of apoptotic and necrotic cells and a reduction in cell migration [64].

The effects of AZD5582 on Fanconi anemia-associated head and neck squamous cell carcinoma (FA-HNSCC) cells have also been investigated. FA-HNSCC is characterized by a very aggressive course, and somatic cells in patients with this disease show hypersensitivity to standard treatments, which often cause very serious side effects, and possibly fatal effects, in patients. Genomic and transcriptomic analyses of FA-HNSCC have shown that the deletion of 18q22.1 and amplification of 11q22.1-q22 represent the most significant changes in terms of both effects on gene expression and frequency of occurrence. Deletion results in the loss of the SMAD4 protein, while amplification results in increased an expression of the YAP1 protein and the cellular inhibitors of apoptosis BIRC2/3 (cIAP1/2). AZD5582 has been shown to result in the decreased cell viability of FA-HNSCC lines with 11q22.2 amplification compared to control cells. A similar effect was observed in a spheroid culture assay in which AZD5582 treatment caused spheroid breakdown or cell separation, as well as an increase in LDH levels, with no concomitant effect on control lines [16].

### 5.2. LCL161

LCL161 is a monovalent SM with a high affinity for cIAP1 and, to a lesser extent, cIAP2 and XIAP [65]. The cytotoxic effect of LCL161 monotherapy on the HNSCC HPV(−) cell lines UM-SCC-1, UM-SCC-74A, Cal27, and UM-SCC-11B has been investigated, as well as the HPV(+) cell lines UM-SCC-47, UD-SCC-2, 93VU147T, and UPC1:SCC090. The IC50 values for all cell lines ranged from 32 to 95 μM, suggesting that LCL161 may not be effective as a monotherapy. However, it was observed that HNSCC HPV(−) cells showed higher sensitivity to LCL161 than HPV(+) cells. The optimal doses and time of treatment of LCL161 cells necessary to reduce IAP levels were also determined. The degradation of cIAP1 and cIAP2 proteins occurred as early as 30 min, and the treatment of LCL161 cells for 2 h led to a dose-dependent reduction in cIAP1 and cIAP2 levels, but not XIAP. No significant changes in the phosphorylation levels of AKT, ERK, and STAT3 were detected in cells exposed to LCL161 for 2 h, suggesting that the JAK/STAT, PI3K/AKT, and MEK/MAPK pathways were not activated. The ability of LCL161 to sensitize cells to radiation therapy was evaluated by treating cells with 100 nM test SM for 2 h, showing that it caused a strong sensitization of HNSCC HPV (−) cells. Compared to radiation alone, the combination treatment significantly increased the population of HNSCC HPV(−) cells in the sub-G1 phase and the cleavage of caspase-3, -7, -8, and -9 and PARP, as well as DNA damage [66].

The effect of LCL161 on HNSCC cell lines (PCI-1, PCI-9, PCI-13, PCI-52, PCI-68) was also tested in combination with the Fas ligand (FasL) [67,68]. The IC50 values obtained for LCL161 alone were in the range of 11–26 μM. LCL161 enhanced the cytotoxic effects of FasL in three of the five cell lines tested (PCI-1, PCI-13, and PCI-68) and sensitized one of the two cell lines resistant to FasL monotherapy (i.e., PCI-9) [67].

In other studies, IC50 values ranged from 19.8 to 64.3 μM depending on the cell line (Detroit 562, FaDu, PCI-1, PCI-9, PCI-13, PCI-52, PCI-68, SCC-9, and SCC-25) [68,69]. Differential sensitivity to FasL was observed among the lines tested, showing that the A-253, SCC-9, and SCC-25 lines were resistant to FasL, while the Detroit 562, FaDu, and HaCaT lines showed a dose-dependent response. The treatment of cells with LCL161 at a concentration equal to IC10 for 72 h sensitized FaDu, Detroit 562, and SCC25 cells and HaCaT lines, inducing a synergistic effect, while the SCC-9 and A-253 lines remained insensitive, and the combination of LCL161 with FasL showed an antagonistic effect. Moreover, LCL161 treatment was shown to induce apoptosis in all cell lines tested [69].

### 5.3. Birinapant

Birinapant (TL32711) is a divalent mimetic that shows a high affinity for the BIR3 domains of cIAP1/2 and XIAP [70,71,72]. It is one of the best studied compounds in the Smac family of mimetics. It has been tested on several HNSCC lines, the vast majority of which were resistant to its monotherapy [68,71,73,74,75]. The compound sensitizes many types of HNSCC cells to other drugs and radiotherapy [71,73,74,75]. Monotherapy with birinapant results in the inhibition of cIAP1 and, to a lesser extent, XIAP expression in HNSCC cells [71,73]. Previous studies and analyses have shown that the increased expression of mRNA and FADD and BIRC2/IAP1 proteins, along with their amplifications on chromosome 11q, contribute to increased cell sensitivity to birinapant [71,74].

The proapoptotic effect of birinapant against HNSCC cells was shown to be enhanced by TRAIL and TNFα [71,75]. Similarly, DNA fragmentation induced by birinapant treatment was significantly enhanced when TNFα was added. Birinapant also showed synergistic effects with docetaxel in vitro [75].

In view of the significant differences in genomic profile between HNSCC HPV(−) and HNSCC HPV(+), we reviewed the characterization of the genomic and expression profiles of TRAIL (TNFSF10) along with its receptors, FADD, BIRC2/3 and XIAP, in HNSCC tissues differing in HPV status. The HNSCC HPV(+) cells rarely showed the amplification and overexpression of FADD but more often exhibited the deletion of the BIRC2/3 loci on chromosome 11q, while also showing the amplification and/or overexpression of the TRAIL or TRAILR gene. Based on these observations, the effect of the anti-TRAILR2 antibody at doses ranging from 100 to 400 ng/mL on cells of the UM-SCC-47 and UPCI-SCC-90 lines has been tested. Monotherapy with the antibody showed no clear effect on cell proliferation, while in combination with birinapant, there was a strong dose-dependent inhibition of proliferation for both cell lines. An even stronger inhibition of proliferation was achieved by treatment with a combination of birinapant, TRAIL, and anti-TRAILR2 antibodies. The combination of birinapant and TRAIL strongly induced cell death by apoptosis, DNA fragmentation, and an increase in the sub-G1 phase—an effect that was enhanced when anti-TRAILR2 antibody was included [74].

The effect of combining birinapant with AZD1775, an inhibitor of WEE1 kinase (a major checkpoint guardian of G2/M and S phases), has been investigated. The combination showed only a slight improvement in efficacy over single birinapant therapy in a panel of HNSCC cell lines (UMSCC-1, UMSCC-11A, UMSCC-11A, UMSCC-22A, UMSCC-46, UMSCC-47, UMSCC-74A, and UPCI-SCC-090). Only the addition of TNFα significantly enhanced the cytotoxic and antiproliferative properties of the combination, while the effect on normal cells was small. Birinapant increased the sub-G1 population in HNSCC HPV(−) cells in monotherapy, while the addition of the WEE1 inhibitor did not significantly affect cell cycle distribution. Only the inclusion of TNFα significantly increased the percentage of the sub-G1 phase. One of the properties of birinapant is its ability to activate NFκB. The combination of birinapant with AZD1775 was shown to abrogate this effect [73]. The inhibition of WEE1 in combination with birinapant has also been shown to sensitize HNSCC cells to radiation therapy-induced apoptosis [71,73].

Studies have confirmed that despite the expression of the Fas receptor on HNSCC cells, FasL monotherapy did not induce significant effects, while birinapant and FasL combination therapy showed synergistic or additive effects in most cell lines tested [68,76].

In addition, HNSCC lines with CASP8 mutation showed higher resistance to radiotherapy relative to HNSCC lines with wild-type CASP8. The knockdown of CASP8 significantly sensitized HNSCC cells (UM-SCC-17A and MOC1) to birinapant-induced necroptosis, and its combination with Z-VAD-FMK and/or radiotherapy maintained RIP3 function [77].

### 5.4. SM-164

SM-164 is a non-peptide, divalent SM that shows a high affinity for the BIR2 and BIR3 domains of IAPs [78]. Studies have been conducted to determine the radiosensitizing properties of SM-164 against four HNSCC cell lines (UMSCC-1, UMSCC-12, UMSCC-17B, and UMSCC-74B). It was shown that all four lines were strongly resistant to SM-164, with IC50 values ranging from 22 to 57 μM. However, it was confirmed that SM-164 radiosensitized two of the four cell lines (UMSCC-1, UMSCC-17B) at a concentration of 200 nM, while at a concentration of 10 nM, the radiosensitizing effect was only noticeable in the UMSCC-1 line. SM-164 induced cIAP1 degradation with no effect on XIAP, leading to the conclusion that the radiosensitizing properties were independent of cIAP1 and XIAP levels. In sensitive cell lines, SM-164 monotherapy increased the activity of caspases-9, -8, and -2, with radiation therapy alone having little effect on caspase activation. The combination of radiotherapy and SM-164 caused an even greater increase in the activity of caspases-9, -8, -2, and -3 and enhanced radiotherapy-induced apoptosis. In SM-164-resistant lines, there was no increase in caspase activity either after treatment with SM-164 alone or in combination with radiotherapy. It was shown that the level of BCL2 in cells correlated with their sensitivity to radiation and the radiosensitizing effect of SM-164. SM-164 also caused an increase in NFκB activity in sensitive cells; the addition of radiotherapy enhanced this effect. The combination therapy also induced an increase in the production of TNFα, high levels of which contributed at least in part to the radiosensitization of UMSCC-1 cells [79].

The response of HNSCC cells to treatment with TRAIL or SM-164 has also been analyzed. Of the nine cell lines tested (HSC3, HSC3M3, HN5, HN30, UMSCC74A, UMSCC74B, H357, UMSCC11B, and UMSCC22B), three (HSC3, HSC3M3, and HN5) showed high sensitivity to the use of SM-164 in monotherapy while remaining resistant to TRAIL. The remaining lines showed the opposite sensitivity pattern. SM-164 induced apoptotic and necroptotic cell death in HNSCC cells. The action of SM-164 on the cells of the sensitive HSC3 line resulted in the cleavage and activation of caspase-10. At the same time, the depletion of caspase-8 had only a slight effect on the degree of cell death, leading to the conclusion that caspase-10 can functionally replace caspase-8 in its absence or low activity. SM-164 also caused the release of cytochrome c and the cleavage and activation of caspase-9, reducing the level of BID protein. However, BID knockout did not affect the effect of SM-164 on the sensitive HSC3 line. This suggests that IAP inhibitor-mediated death is BID-independent. SM-164 caused the depletion of cIAP1, downregulation of XIAP, and activation of caspase-3 in HSC3M3 cells. The degradation of cIAP1 occurred in lines sensitive to SM-164 monotherapy, as well as resistant ones, suggesting that the difference in response between different cells is not solely dependent on cIAP1 levels. The level of XIAP in cells did not correlate with the sensitivity of cells to SM-164. SM-164 induced the autocrine secretion of TNFα, whose level of secretion correlated with the sensitivity of cells to treatment with this SM. Sensitive cells showed high NF-κB activity, which increased after SM-164 treatment [80].

TRAIL combination therapy with SM-164 resulted in the strong growth arrest of nasopharyngeal carcinoma S18 cells, which showed resistance to TRAIL. Both monotherapy and TRAIL combination therapy reduced the fractions of so-called side population cells (SP). SP cells are identified by their ability to remove the fluorescent dye Hoechst 33342. According to previous reports, this feature may be a marker of cancer stem cells. The ability of the cells to form colonies and spheres was also observed. Treatment of the cells with SM-164 resulted in a decrease in the levels of cIAP1, XIAP, pro-PARP, and procaspase-3 and an increase in the levels of active PARP and caspase-3—an effect that was enhanced when TRAIL was added. The combination therapy led to a strong induction of apoptosis in the cells tested [81].

### 5.5. BV6

BV6 is a dual SM showing an affinity for cIAP1, cIAP2, and XIAP. One study that evaluated the efficacy of BV6 monotherapy using a panel of 10 cell lines (PCI-1, PCI-9, PCI-13, PCI-52, PCI-68, Detroit 562, FaDu, SCC-9, SCC-25, and HaCaT) showed concentration-dependent cytotoxic effects for all cell lines tested, with IC50 ranging from 1.2 to 8.7 μM depending on the cell line. In nine cell lines (excluding SCC-9) exposed to BV6 at a concentration equal to IC10 for 48 h, a significant increase in the late apoptotic cell population was observed. CIAP1 depletion occurred in all cell lines tested, excluding HaCaT and SCC9 cells. In the SCC25, Detroit 562, PCI-1, PCI-9, PCI-13, and PCI-52 cell lines, cIAP2 was degraded, and the XIAP was also degraded in the same lines, except for PCI-1. Combination therapy in the form of a combination of BV6 at IC10 and FasL showed synergistic effects and resulted in a significant increase in cytotoxicity compared to FasL monotherapy in 8 of the 10 cell lines tested. SCC-9 and PCI-52 cells remained resistant to both FasL monotherapy and combination treatment [68].

BV6’s ability to sensitize cells to other therapeutics was also confirmed in a study using cisplatin-sensitive (SCC4) and -resistant (SCC4cisR) tongue cancer cell lines. In monotherapy, BV6 showed relatively potent cytotoxic effects, with an IC50 of 2.0 μM for both lines tested. The compound decreased the levels of cIAP1, XIAP, and livin in a dose-dependent manner. It was also shown that XIAP degradation increased with the increasing treatment time of cells with this SM. BV6 caused the sensitization of cells to cisplatin-induced apoptosis, including cells originally resistant to platinum treatment. A synergistic effect was observed between cisplatin and BV6 [82].

### 5.6. ASTX660

ASTX660 is a potent, non-peptidomimetic dual antagonist of cIAPs and XIAPs. A series of studies was conducted to evaluate the effects of ASTX660 on mouse oral cancer cells of the MOC1, MOC2, and MOC22 lines, as well as its sensitizing effects on TNFα, TRAIL, and FasL treatment. It was shown that all tested lines were resistant to ASTX660 in monotherapy at concentrations up to 10 μM. However, ASTX660 sensitized MOC22 cells to TNFα and sensitized MOC1 cells to TNFα, TRAIL, FasL, and cisplatin. ASTX660-treated cells showed a reduction in cIAP1 levels without concomitant changes in cIAP2 and XIAP levels. ASTX660 sensitized the tumor cells of the MOC1ova line to specific cytotoxic T cells. ASTX660 at concentrations < 10 µM had no concomitant effect on the viability and proliferative capacity of T lymphocytes. The increased sensitivity of MOC1ova cells to T-lymphocyte-mediated death was mediated by ligands of the perforin/granzyme B superfamily and TNFR (TNFα, TRAIL, and FasL) [83].

In studies using a panel of 11 HNSCC cell lines, including six HPV(+) lines (i.e., UM-SCC-47, UMSCC-104, UPCI-SCC-90, UPCI-SCC-152, UD-SCC-2, and 93VU147T) and five HPV(−) lines (i.e., UM-SCC-11B, UM-SCC-22B, UM-SCC-38, UM-SCC-46, and UM-SCC-74A), the cells of the UM-SCC-46 line proved most sensitive to ASTX660, while the other test lines were considered resistant to this IAP inhibitor, as the IC50 value exceeded 1 μM (Table 1). The combination of ASTX660 with TRAIL and TNFα resulted in the sensitization of HPV(−) lines showing increased expression of FADD and/or cIAP1, XIAP, and wild-type TP53, as well as several HPV(+) lines that did not show increased FADD, cIAP1, and XIAP copy numbers. The combination of ASTX660 with death ligands showed a synergistic effect. However, in the UM-SCC-38 line, the combination with TRAIL showed an additive effect, and antagonistic effects were seen in the UDSCC-2 lines: 93VU147T for TRAIL and 93VU147T for TNFα. Association therapy of ASTX660 with death ligands was shown to work by apoptosis and/or necroptosis. ASTX660 also caused a significant degradation of the cIAP1 protein both in monotherapy and in combination with TNFα. During the study, there was a compensatory effect of increasing cIAP2 levels. ASTX660 did not cause the degradation of XIAP and caused a decrease in caspase-8 and caspase-3 levels, suggesting their cleavage and activation. There was also an increase or decrease in MLKL levels depending on the line. In the tested HPV-positive cell lines, under the influence of ASTX660 and TNFα, there was a significant increase in the expression of TP53 protein and p21 protein. A decrease in MDM2, a negative regulator of TP53, was also observed. The role of TP53 was confirmed by its knockdown, whereby the HPV(+) cells tested became more resistant to ASTX660 combination therapy with TRAIL or TNFα [84].

Studies have shown that ASTX660 combined with TNFα can induce immunogenic cell death through the release of damage-associated molecular patterns (DAMPs) by cancer cells, which include the release of high-mobility group 1 proteins (HMGB1) and adenosine 5-triphosphate (ATP) or CXCL10 from the intracellular environment, as well as the surface exposure of the heat shock proteins HSP70 and HSP90 and the expression of calreticulin (CRT) on the surface of cancer cells. DAMP release leads to an increase in cellular immunogenicity and the enhancement of the immune response through the activation of CD8+ dendritic cells. In one study, combination therapy with ASTX660 and TNFα led to an increase in surface HSP70 and CRT and enhanced HMGB1 release in HPV(−) UMSCC-46 and HPV(+) UMSCC-47 cell lines. The combination of ASTX660 with TNFα resulted in an increase in MHC class I (HLA-A,-B,- C) expression levels in all human cell lines tested (UMSCC-11B, UMSCC-46, UMSCC-47, and UMSCC-74A). Other components of the so-called antigen presentation machinery, such as ERp57, LMP2, TAP1, TAP2, and CRT, showed differential changes depending on the cell line [64]. ASTX660 has been found to have no cytotoxic effects on three mouse cancer lines—MOC1, MOC2, and MEER (a model developed to express E6 and E7 HPV proteins)—and in combination with TNFα caused an increase in the surface expression of CRT/HSP70 only in the MOC1 line. An increase was also found in the population of cells with low levels of intracellular HMGB1 in both the MOC1 and MEER cells. ASTX660 enhanced the killing of tumor cells by tumor-infiltrating lymphocytes (TILs) in both TIL monotherapy-sensitive lines and in resistant cells (MEER) by sensitizing them [85].

### 5.7. Xevinapant

Xevinapant, also called Debio 1143 or AT-406, is a monovalent SM that binds to cIAPs and XIAPs. In a study that used a panel of seven HNSCC cell lines (HPV), i.e., HSC4, Detroit 562, RPMI-2650, CLS254, Cal33 and HPV (+), i.e., UM-SCC-47 and UD-SCC-2) and two control lines (SBLF9 and 01-GI-SBL), xevinapant at a concentration of 5 μM showed no significant differences in cell proliferative activity, while at concentrations above 8.4 μM, it strongly attenuated the growth rate. At concentrations of 13.3 and 16.7 µM, xevinapant in monotherapy, as well as in combination with irradiation, showed cytotoxic effects on almost all cell lines. At a concentration of 8.4 µM, xevinapant inhibited cell proliferation. In the colony formation assay, xevinapant in monotherapy significantly reduced the survival fraction, and the combination with irradiation amplified the effects relative to treatment with radiotherapy alone. Cell death from treatment with xevinapant and its combination with radiation therapy occurred through both apoptosis and necroptosis. The population of apoptotic cells increased as the concentration used increased. Xevinapant caused the formation of double-stranded DNA breaks, which the cells were unable to repair after 48 h. These effects were observed in all tested lines except UD-SCC-2, where a reduction in γH2AX foci was observed. In most cells, the cIAP1 and XIAP levels increased after treatment with xevinapant alone or radiotherapy alone. The combination treatment resulted in a decrease in cIAP1 levels in all HNSCC cells, while XIAP levels remained elevated in most lines [86].

In another study, xevinapant showed moderate cytotoxicity against S18 and S26 nasopharyngeal cancer cells. The S18 line showed low sensitivity to TRAIL, and the association of TRAIL and xevinapant resulted in strong inhibition of S18 line cells. Xevinapant reduced the proportion of SP cells in S18 cells in monotherapy. The inclusion of TRAIL enhanced this effect. The compound also inhibited the ability of cells to form colonies and spheres, especially in combination therapy. Combination therapy caused induction of apoptosis in the S18 and S26 cell lines tested. A decrease in the levels of pro-PARP and procaspase-3 and an increase in the levels of cleaved caspase-3 and PARP were observed in the cell lines. cIAP1 was degraded in cells treated with xevinapant alone as well as in combination with TRAIL [81].

A study conducted on six HNSCC cell lines (SQ20B, FaDu, Cal27, RPMI-2650, SCC-4, and SCC-15) showed varying cytotoxic effects of xevinapant. Two lines (Cal27 and SCC-4) proved relatively sensitive, three moderately so (RPMI-2650, SCC-15, and FaDu), and one (SQ20B) showed low sensitivity to the effects of xevinapant. In combination with radiotherapy (combination therapy), xevinapant had a much stronger effect. Only cells of the SCC15 line showed high resistance to both radiation monotherapy and combination therapy. The highest treatment efficacy with combination therapy was achieved when SM was administered continuously or added 24 h after irradiation and applied for 10 days. Xevinapant caused cIAP1 degradation in all cell lines tested. Combination therapy caused an increase in the apoptotic cell population, stronger than with irradiation alone; an increase in the number of pyknotic nuclei in cells; and an increase in TNFα mRNA expression. In SQ20B cells, both xevinapant alone and the combination therapy caused an increase in caspase-3, while in FaDu cells, the effect was small. The use of a pan-caspase inhibitor and an anti-TNFα neutralizing antibody reversed the radiosensitizing properties of xevinapant, leading to the conclusion that TNFα and caspases are involved in sensitizing cells to radiotherapy [87].

In a study conducted on CC4 and SCC4cisR tongue cancer cell lines, xevinapant showed minimal cytotoxic properties. At concentrations of 5, 10, and 20 µM after 48 h incubation, xevinapant induced degradation of cIAP1 protein, while having no effect on XIAP or livin levels. When combined with cisplatin, there was no increase in apoptosis induced by the chemotherapeutic agent alone [82].

### 5.8. APG-1387

APG-1387 is a potent Smac mimetic that shows an affinity for cIAP1, cIAP2, and XIAP. In studies on human nasopharyngeal carcinoma S-18 and S-26 cells, it strongly induced the degradation of cIAP1/2 and XIAP, reduced the cells’ ability to migrate in a dose-dependent manner, reduced the SP fraction, and downregulated Sox2 protein—an important regulator of embryonic stem cell fate. At low concentrations, APG-1387 strongly inhibited the formation of cell spheres of the S-18 lineage. The application of APG-1387 sensitized nasopharyngeal cancer cells to treatment with 5-fluorouracil (5-FU) and cisplatin. APG-1387 also caused the potent cleavage of poly(ADP-ribose) polymerase (PARP) in both monotherapy and combination therapy [88].

### 5.9. Embelin

One study showed that embelin (a non-peptide, potent XIAP inhibitor) prevents cellular resistance to TRAIL induced by latent membrane protein 1 (LMP1), in which the XIAP is involved. The treatment of CNE-1-LMP1 cells (i.e., CNE-1 lineage cells transfected with pGL6-LMP1 to increase LMP1 protein expression) with embelin led to a concentration-dependent decrease in XIAP expression levels. Combination therapy of embelin and TRAIL resulted in increased levels of apoptosis relative to TRAIL monotherapy. The effects on the EBV(+) cell line C666-1 were investigated and showed that the combination of TRAIL and embelin enhanced the apoptotic effect relative to TRAIL monotherapy and decreased XIAP levels in control cells but did not enhance TRAIL-induced apoptosis [89]. Embelin showed synergistic effects with cisplatin and enhanced apoptosis and cIAP1 and XIAP degradation induced by this chemotherapeutic in cells of the SCC4cisR line. Embelin did not cause a degradation of IAPs in monotherapy, and in combination therapy, it neither enhanced this process nor led to the promotion of chemotherapy-induced apoptosis [82].

**Table 1 pharmaceuticals-17-01308-t001:** The table presents the IC50 results of IAP inhibitors obtained from in vitro studies.

Compound	Form of Treatment	Cell Line	IC_50_	Time	Source
Birinapant	Monotherapy	UM-SCC-1	>1000 nM	72 h	[71,75]
		UM-SCC-6	>1000 nM		
		UM-SCC-9	>1000 nM		
		UM-SCC-11A	>1000 nM		
		UM-SCC-11B	>1000 nM		
		UM-SCC-22A	>1000 nM		
		UM-SCC-22B	>1000 nM		
		UM-SCC-38	>1000 nM		
		UM-SCC-46	10.7 nM		
		UM-SCC-74A	>1000 nM		
		UM-SCC-74B	>1000 nM		
		UM-SCC-47	>1000 nM		
		93VU147T	>5000 nM	72 h	[74]
		UM-SCC-47	>5000 nM		
		UM-SCC-104	>5000 nM		
		UM-SCC-105	>5000 nM		
		UPCI-SCC-90	>5000 nM		
		UPCI-SCC-152	>5000 nM		
		UPCI-SCC-154	>5000 nM		
		UD-SCC-2	>5000 nM		
		93VU147T	>5000 nM	120 h	
		UM-SCC-47	500 nM		
		UM-SCC-104	>5000 nM		
		UM-SCC-105	631 nM		
		UPCI-SCC-90	>5000 nM		
		UPCI-SCC-152	>5000 nM		
		UPCI-SCC-154	>5000 nM		
		UD-SCC-2	>5000 nM		
		Detroit 562	>79.5 µM	72 h	[76]
		FaDu	>58.3 µM		
		PCI-1	41.0 µM		
		PCI-9	21.4 µM		
		PCI-13	31.1 µM		
		PCI-52	>67 µM		
		PCI-68	>56.1 µM		
		SCC-9	>19.0 µM		
		SCC-25	>4.9 µM		
	Birinapant + 20 ng/mL TNFα	UM-SCC-1	45.3 nM	72 h	[71,75]
		UM-SCC-6	2.98 nM		
		UM-SCC-9	>1000 nM		
		UM-SCC-11A	47.1 nM		
		UM-SCC-11B	41.7 nM		
		UM-SCC-22A	1.88 nM		
		UM-SCC-22b	>1000 nM		
		UM-SCC-38	>1000 nM		
		UM-SCC-46	0.72 nM		
		UM-SCC-74A	8.78 nM		
		UM-SCC-74B	0.1 nM		
		UM-SCC-47	>1000 nM		
		93VU147T	200 nM	72 h	[74]
		UM-SCC-47	20 nM		
		UM-SCC-104	2.5 nM		
		UM-SCC-105	>5000 nM		
		UPCI-SCC-90	5.6 nM		
		UPCI-SCC-152	1.1 nM		
		UPCI-SCC-154	>5000 nM		
		UD-SCC-2	794 nM		
		93VU147T	177 nM	120 h	
		UM-SCC-47	1.6 nM		
		UM-SCC-104	0.3 nM		
		UM-SCC-105	661 nM		
		UPCI-SCC-90	1 nM		
		UPCI-SCC-152	0.4 nM		
		UPCI-SCC-154	>5000 nM		
		UD-SCC-2	1995 nM		
	Birinapant + 50 ng/mL TRAIL	UM-SCC-1	17.72 nM	72 h	[71,75]
		UM-SCC-6	0.29 nM		
		UM-SCC-9	> 1000 nM		
		UM-SCC-11A	1.59 nM		
		UM-SCC-11B	39.9 nM		
		UM-SCC-22A	>1000 nM		
		UM-SCC-22B	>1000 nM		
		UM-SCC-38	>1000 nM		
		UM-SCC-46	0.57 nM		
		UM-SCC-74A	65.4 nM		
		UM-SCC-74B	0.27 nM		
		UM-SCC-47	160 nM		
		93VU147T	100 nM	72 h	[74]
		UM-SCC-47	32 nM		
		UM-SCC-104	>5000 nM		
		UM-SCC-105	2 nM		
		UPCI-SCC-90	>5000 nM		
		UPCI-SCC-152	8.9 nM		
		UPCI-SCC-154	0.4 nM		
		UD-SCC-2	>5000 nM		
		93VU147T	126 nM	120 h	
		UM-SCC-47	28 nM		
		UM-SCC-104	>5000 nM		
		UM-SCC-105	3.2 nM		
		UPCI-SCC-90	25 nM		
		UPCI-SCC-152	10 nM		
		UPCI-SCC-154	0.6 nM		
		UD-SCC-2	>5000 nM		
	FasL + IC10 Birinapant	Detroit 562	1.3 ng/mL	72 h	[76]
		FaDu	7.8 ng/mL		
		PCI-1	1.5 ng/mL		
		PCI-9	14.0 ng/mL		
		PCI-13	3.3 ng/mL		
		PCI-52	-		
		PCI-68	27.4 ng/mL		
		SCC-9	-		
		SCC-25	12.9 ng/mL		
AZD5582	Monotherapy	Cal27	4.21 µM	72 h	[64]
		SCC25	0.54 µM		
		FaDu	3.02 µM		
		VU974-T	3.9 nM	96 h	[16]
		VU1604	3.9 nM		
		VU1365-T	250 nM		
		CCH-FAHNSCC-1	>1000 nM		
		CCH-FAHNSCC-2	>1000 nM		
		VU1131-T	>1000 nM		
	AZD5582 + irradiation 2 Gy	Cal27	3.03 µM	72 h	[64]
		SCC25	0.43 µM		
		FaDu	2.79 µM		
	AZD5582 + irradiation 4 Gy	Cal27	2.70 µM	72 h	[64]
		SCC25	0.27 µM		
		FaDu	3.43 µM		
	AZD5582 + irradiation 8 Gy	Cal27	1.57 µM	72 h	[64]
		SCC25	0.17 µM		
		FaDu	2.67 µM		
LCL161	Monotherapy	Cal27, UM-SCC-1, UM-SCC-74A, UM-SCC-11B, UM-SCC-47, UD-SCC-2, UPC1-SCC-090, 93VU147T	32–95 µM	72 h	[66]
		PCI-1	12/30.6 µM	72 h	[67,68]
		PCI-9	26/36.1 µM		
		PCI-13	11/41.1 µM		
		PCI-52	13/40.2 µM		
		PCI-68	17/49.3 µM		
		Detroit 562	48.7 µM		[67]
		FaDu	61.2 µM		
		SCC-9	21.6 µM		
		SCC-25	19.8 µM		
		A-253	64.3 µM	72 h	[69]
	FasL + IC10 LCL161	Detroit 562	1.7 ng/mL	72 h	[68]
		FaDu	5.2 ng/mL		
		PCI-1	1.8 ng/mL		
		PCI-9	4.9 ng/mL		
		PCI-13	1.7 ng/mL		
		PCI-52	-		
		PCI-68	24.8 ng/mL		
		SCC-9	-		
		SCC-25	17 ng/mL		
		A-253	-		[69]
SM-164	Monotherapy	UMSCC-1	32.1 µM	24 h	[79]
		UMSCC-12	55.2 µM		
		UMSCC-17B	22.3 µM		
		UMSCC-74B	56.7 µM		
		HSC3	0.03 nM	72 h	[80]
		HSC3M3	0.47 nM		
		HN5	0.25 nM		
		S18	5.07 µM	48 h	[81]
		S26	1.37 µM		
BV6	Monotherapy	Detroit 562	8.7 µM	72 h	[68]
		FaDu	3.6 µM		
		PCI-1	4.5 µM		
		PCI-9	3.7 µM		
		PCI-13	5.6 µM		
		PCI-52	6.6 µM		
		PCI-68	3.5 µM		
		SCC-9	1.6 µM		
		SCC-25	1.2 µM		
		SCC-4	2.0 ± 0.5 μM		[82]
		SCC-4cisR	2.0 ± 0.2 μM		
	FasL + IC10 BV6	Detroit 562	1.3 ng/mL	72 h	[68]
		FaDu	2.4 ng/mL		
		PCI-1	1.9 ng/mL		
		PCI-9	13.9 ng/mL		
		PCI-13	2.3 ng/mL		
		PCI-52	-		
		PCI-68	26.3 ng/mL		
		SCC-9	-		
		SCC-25	10.3 ng/ml		
ASTX660	Monotherapy	UM-SCC-11B	>1000 nM	72 h	[84]
		UM-SCC-22B	>1000 nM		
		UM-SCC-38	>1000 nM		
		UM-SCC-46	4.3 nM		
		UM-SCC-74A	>1000 nM		
		UM-SCC-47	>1000 nM		
		UM-SCC-104	>1000 nM		
		UPCI-SCC-90	>1000 nM		
		UPCI-SCC-152	>1000 nM		
		UD-SCC-2	>1000 nM		
		93-VU-147T	>1000 nM		
	ASTX660 + 20 ng/mL TNFα	UM-SCC-11B	68.2 nM	72 h	[84]
		UM-SCC-22B	0.1 nM		
		UM-SCC-38	>1000 nM		
		UM-SCC-46	0.9 nM		
		UM-SCC-74A	4.1 nM		
		UM-SCC-47	285 nM		
		UM-SCC-104	>1000 nM		
		UPCI-SCC-90	14.3 nM		
		UPCI-SCC-152	43.9 nM		
		UD-SCC-2	>1000 nM		
		93-VU-147T	>1000 nM		
	ASTX660 + 50 ng/mL TRAIL	UM-SCC-11B	27.7 nM	72 h	[84]
		UM-SCC-22B	>1000 nM		
		UM-SCC-38	>1000 nM		
		UM-SCC-46	0.5 nM		
		UM-SCC-74A	0.3 nM		
		UM-SCC-47	18.9 nM		
		UM-SCC-104	>1000 nM		
		UPCI-SCC-90	21.2 nM		
		UPCI-SCC-152	24.3 nM		
		UD-SCC-2	>1000 nM		
		93-VU-147T	>1000 nM		
Xevinapant	Monotherapy	S18	233.9 µM	48 h	[81]
		S26	20.92 µM		
		SQ20B	16.71 µM	Nd.	[87]
		FaDu	2.45 µM		
		Cal27	1.16 µM		
		RPMI-2650	1.69 µM		
		SCC-4	0.41 µM		
		SCC-15	1.21 µM		
		SCC-4	>100 μM	72 h	[82]
		SCC-4cisR	>100 μM		
Embelin	Monotherapy	SCC-4	>100 μM	72 h	[82]
		SCC-4cisR	>100 μM		

## 6. Animal Studies

### 6.1. Birinapant

To evaluate birinapant activity, studies have been conducted using UM-SCC-46 cells with an increased expression of FADD and a normal expression of BIRC2 and UM-SCC-11B cells in mouse xenograft models. It was shown that birinapant had a significant inhibitory effect on tumor growth in both models. In the case of the UM-SCC-46 heterotransplant model, a significant difference in mean tumor volumes was observed in the control animals compared to the treated animals, and an improvement in median survival was observed. For UM-SCC-11B, prolonged tumor reduction was observed during and after treatment with birinapant at 30 mg/kg or 15 mg/kg/day, and prolonged survival was observed with both treatment regimens. The doses used were well tolerated by the animals, and all mice maintained normal body condition scores and body weight during treatment. Birinapant was shown to significantly reduce cIAP1, cIAP2, and XIAP levels during long-term treatment. The necroptosis marker MLKL was significantly elevated in tumors during the treatment period. No significant differences were found in staining for markers of proliferation (Ki67) or apoptosis (TUNEL) in tumor samples at any time point, consistent with the primarily MLKL/necroptosis-dependent inhibitory effects observed in this model.

It was also confirmed that treatment with either birinapant or radiation significantly delayed tumor growth and improved survival. Interestingly, birinapant combined with radiation cured the animals with xenografts derived from UM-SCC-46. TNFα expression in tumors was increased by the combination of birinapant and radiation therapy compared to each alone. Birinapant in combination with radiation was observed to strongly induce endogenous TNFα in tumors in vivo, and the combination of TNFα (2 ng/mL) with birinapant was confirmed to significantly increase radiation sensitivity at 3 Gy compared to single agents. The combination of birinapant and irradiation or irradiation alone also significantly inhibited tumor growth in UM-SCC-11B xenografts overexpressing FADD and BIRC2, but not in UM-SCC-1 xenografts showing weaker growth and FADD/BIRC2 copy expression and heterozygous CASP8 mutation [71].

The combination of birinapant and docetaxel in HNSCC was also evaluated. After nine days of treatment with birinapant, there was a significant difference in tumor volume compared to the controls. Surprisingly, docetaxel showed no effect in vivo despite the sensitivity of the cells in vitro. The combination therapy group showed significantly reduced tumor volumes compared to the control group receiving birinapant as a monotherapy. Birinapant used in monotherapy significantly improved survival compared to the control group [75].

The loss of Casp8 increases sensitivity to birinapant and enhances the radiosensitizing effect of birinapant in MOC1 cells in vitro. Therefore, we evaluated the efficacy of birinapant and combination therapy with radiation in the absence or presence of Casp8 knockdown in in vivo studies. Radiation treatment significantly inhibited tumor growth in vivo and improved survival in both the control and Casp8 knockdown animal groups. However, birinapant proved effective in retarding tumor growth only when combined with Casp8 knockdown, accompanied by a significant survival benefit. The combination of radiation and birinapant further reduced tumor growth in vivo and provided a survival benefit in both the control and Casp8 knockdown groups. The results suggest that a loss of CASP8 sensitizes HNSCC to birinapant and enhances its in vivo radiation-sensitizing effects [77].

### 6.2. LCL161

LCL161 has also been evaluated for its ability to sensitize HNSCC in xenografts from nude mice. LCL161 had no effect on tumor growth, while radiation therapy caused partial growth retardation. However, LCL161 combined with radiotherapy led to a sustained regression of xenografts of both Cal27 and FaDu cells, with most Cal27 tumors showing no evidence of tumor recurrence. The treatment was well tolerated, although mice receiving the combination treatment lost weight slightly but fully recovered within 1–2 weeks. The study showed the degradation of cIAP1 by LCL161 in monotherapy and in combination therapy. Radiation alone did not induce cIAP1 degradation or the cleavage of caspase-3, -7, -8, and -9 and PARP. In contrast, a combination treatment of LCL161 with radiation induced the strong cleavage of caspase-3, -7, -8, and -9 and PARP, suggesting that the activation of apoptosis is the main mechanism for tumor growth regression and sustained response [66].

### 6.3. APG-1387

A xenograft tumor model was used to test whether APG-1387 can affect tumor growth in combination with cisplatin or 5-FU in vivo. It was shown that the tumor volume in mice treated with APG-1387 (10 mg/kg) in combination with cisplatin or 5-FU was significantly smaller than in mice treated with a single agent. Importantly, no toxicity was observed in the mice during treatment, and the body weight of the mice did not change. To define the properties of S-18 stem cells after combined treatment with APG-1387 and conventional in vivo therapy, cells derived from a primary xenograft were implanted into non-obese diabetic/severe combined immunodeficiency mice. Tumor cells derived from mice treated with APG-1387 resulted in slower tumor growth. However, tumor cells from mice treated with cisplatin or 5-Fu showed rapid tumor regrowth. Most notably, the tumor regrowth of cells in mice receiving APG-1387 in combination with cisplatin treatment was the slowest among the study groups. APG-1387 in combination with conventional chemotherapeutics showed a significant synergistic effect in inhibiting nasopharyngeal carcinoma cells in vivo [88].

### 6.4. SM-164

A study was conducted to determine the ability to sensitize cells to radiation after SM-164 treatment in vivo using the UMSCC-1 heterotransplant model. It was shown that SM-164 administration eliminated cIAP-1, starting 3 h after treatment. As expected, radiation alone had no effect on cIAP1. The combination of SM-164 and radiation therapy also eliminated cIAP-1, and cIAP-1 did not return to baseline levels in any tumor. The results showed that SM-164 is effective against cIAP1 and suggested that radiation should be delivered 2–3 h after SM-164 administration, when cIAP1 is undetectable. The combination of SM-164 and radiation therapy resulted in the inhibition of tumor growth. The combination treatment was well tolerated by the animals with minimal weight loss. The results confirm that SM-164 sensitizes UMSCC-1 head and neck cancer cells to radiation and acts as a new class of radiosensitizers [79].

### 6.5. ASTX660

To determine whether ASTX660 can enhance radiation-induced immune cell death in vivo, studies have been conducted using mouse oral cancer 1 (MOC1) and MEER, a mouse model designed to express HPV E6 and E7 oncoproteins. Mice were inoculated subcutaneously with MOC1 or MEER cells treated in vitro with mitoxantrone (positive control), ASTX660 + TNFα, radiation, or ASTX660 + radiation.

It was shown that MOC1 cells killed by radiation induced a strong immune response, with a rejection of tumor formation in 50% of mice. Cells killed by combination therapy (ASTX660 + radiation) induced an even stronger immune response, with a rejection of tumor formation in up to 72% of mice. MEER cells treated with radiation or combination therapy (ASTX660 + radiation) induced a similar immune response, with the combination showing no difference from the tumor rejection rate of 80% observed with radiation alone. However, an additional delay in tumor growth was observed in the group receiving combination therapy (ASTX660 + radiation). Treatment with ASTX660 + TNFα did not kill cells of the MOC1 or MEER lineage to any significant extent, leading to tumor engraftment in mice. The results indicate that ASTX660 can promote cell death in vivo, but its effects are best when combined with radiation [85].

The researchers also used the heterotransplantation of HNSCC HPV(−) UM-SCC-46 and HPV(+) UPCI-SCC-90 cancers to evaluate the efficacy of ASTX660 in monotherapy and in combination therapy with radiation. It was confirmed that in untreated UM-SCC-46 tumor-bearing mice, the tumors grew rapidly, with a median survival of 42 days. ASTX660 administration slightly delayed tumor growth and provided a significant survival benefit (median survival of 53 days), while radiation slightly delayed tumor growth but significantly prolonged survival compared to both the control group and the group receiving ASTX660 in monotherapy (70 days). The combination therapy (ASTX660 + radiation) was shown to significantly delay tumor growth, and all treated mice survived the full 100 days, confirming that the combination therapy provided a significant survival benefit compared to all other groups.

The role of TNFα in the mechanism by which ASTX660 interacts with radiation to delay the growth of UM-SCC-46 tumors was also investigated. Mice were treated with combination therapy (ASTX660 + radiation) under a TNFα blockade. The control mice were shown to experience rapid tumor growth, with a median survival of 42 days. When the TNFα-deficient mice were given the combination treatment, tumor growth was delayed compared to the control mice, and the median survival was 74 days vs. 100 days in mice without the TNFα blockade, confirming the key role of TNFα in the antitumor effect of the combination therapy (ASTX660 + radiation).

The efficacy of ASTX660 in monotherapy and combination therapy has also been analyzed in an HPV(+) HNSCC UPCI-SCC-90 heterotransplant model. Control tumors in untreated mice grew rapidly after reaching a critical mass, with a median survival of 55 days. ASTX660 alone moderately delayed tumor growth. In contrast, radiation alone significantly delayed tumor growth and extended median survival to 83 days. Combination therapy (ASTX660 + radiation) significantly delayed tumor growth and increased survival, with most mice surviving the full 100 days after inoculation. These results suggest that combination therapy (ASTX660 + radiation) may be effective in treating HPV(−) as well as HPV(+) HNSCC [84].

The effectiveness of ASTX660 has also been evaluated using a syngeneic model of MOC1 in vivo in monotherapy and in combination with immune checkpoint blockade and chemotherapy. It was shown that in MOC1 mice, ASTX660 alone resulted in a small increase in median survival of three days. Each of the combination therapies (ASTX660 + cisplatin, ASTX660 + anti-PD-1, and ASTX660+cisplatin + anti-PD-1) significantly reduced tumor growth and improved overall survival. Compared to the controls, in which median survival was 41 days, each combination therapy regimen provided a significant survival benefit, as follows: ASTX660 + cisplatin (68 days), ASTX660 + Anty-PD-1 (50 days), and ASTX660 + CDDP + Anty-PD-1 (65 days). ASTX660 has been shown to not add much to combination therapy, but it may be a less-toxic substitute for cisplatin or anti-PD-1.

The combination of anti-PD-1 and ASTX660 has also been shown to significantly increase the intratumor CD8+ T cell count, and this effect did not disappear when cisplatin was added. The number of CD11B + CD11C + dendritic cells with B7-1 expression (CD80) increased in a subgroup of mice tumors treated with the triple combination but without statistical significance. It was shown that ASTX660 (median survival of 72 days) and radiation alone (82 days) significantly delayed tumor growth and improved survival compared to the control group (65 days). Combination therapy (ASTX660 + radiation) significantly delayed tumor growth and improved survival in mice. Triple combination therapy (ASTX660 + radiation + anti-PD-1) caused the greatest delay in tumor growth, and most mice rejected tumors, suggesting that this combination is most effective in promoting the antitumor immune response [83].

### 6.6. Xevinapant

To demonstrate the efficacy of xevinapant, mouse experimental models have been used, i.e., resistant (SQ20B) and radiation-sensitive (FaDu). Mice with SQ20B xenografts were treated for 14 consecutive days with Debio 1143 and concomitant radiation therapy, while FaDu mice received Debio 1143 for 2 or 3 weeks and concomitant radiation therapy. In the mice in the radiation-resistant model, oral administration of Debio 1143 (100 mg/kg) did not show any antitumor effect; however, the administration of Debio 1143 at doses of 10, 30, and 100 mg/kg/24 h for 14 days in combination with radiation therapy (2 Gy per day) dose-dependently enhanced the antitumor effect of radiation (Figure 7). Debio 1143 at doses of 30 and 100 mg/kg in combination with radiotherapy was highly effective in inhibiting tumor growth while showing no toxic effects on animals. In addition, it was shown that combined therapy (Debio 1143 and radiotherapy) translated into increased survival compared to mice treated with radiotherapy alone. In the FaDu model, three-week treatment with Debio 1143 (100 mg/kg 5 days a week) combined with 2 Gy of radiotherapy was more effective than treatment for two weeks. Importantly, the three-week treatment induced a complete response in 80% of the mice, while radiotherapy alone or Debio 1143 did not cure any animals [87].

## 7. Clinical Studies

Studies on IAP antagonists have shown promising results in the treatment of head and neck cancer, particularly when combined with radiotherapy, as evidenced by numerous clinical trials (Table 2). In our article, in addition to preclinical studies, we decided to conduct a review of clinical trials concerning the use of new IAP inhibitors in the treatment of head and neck cancer. Most clinical trials are initially conducted on participants with a confirmed recurrence of the disease or distant metastases. After demonstrating treatment efficacy, these compounds are tested in cases of primary disease. However, due to the unique mechanism of action of IAP inhibitors and their ability to overcome chemotherapy resistance, they are expected to be particularly effective in extending disease-free intervals and improving cure rates in at least some selected chemotherapy and/or radiotherapy protocols.

IAP inhibitors, such as birinapant, xevinapant, and tolinapant, are currently being investigated in phase I, II, or III trials. The ongoing studies primarily focus on combining IAP inhibitors with radiotherapy, immunotherapy, or chemotherapy, as well as various combinations of standard approved therapies [90].

In the completed NCT02022098 study, an oral formulation containing the Smac mimetic Debio 1143 (Xevinapant) was used to regulate cIAP1 levels. It was indicated that enhancing cisplatin chemotherapy with this mimetic in the treatment of squamous cell carcinoma of the head and neck resulted in increased therapeutic efficacy [91,92,93,94]. The study employed a daily dose of 200 mg Xevinapant, with the possibility of dose reduction down to 100 mg depending on the patient’s response [93]. Interestingly, the group treated with the combination therapy was compared to a placebo group, and a slight increase in manageable adverse events was observed [91,92,94]. However, researchers reported that Xevinapant, in combination with radiotherapy, may exacerbate epithelial cell death, potentially leading to mucositis or dysphagia [93]. It is worth noting that the results of this study are not representative for HPV-positive cancer patients due to an insufficient recruitment of patients with this cancer type [91,92].

Also, clinical studies conducted on SMs in the treatment of other cancers reveal promising prospects for their use, further strengthening the belief that they could be valuable allies in the fight against HNSCC. Currently completed clinical trials primarily focus on solid tumors of various types [65,91,92,93,94,95,96,97,98,99,100,101,102]. It has been indicated that Smac mimetics, by reducing the levels of cIAP1/2 and/or XIAP, can effectively enhance the efficacy of conventional chemotherapy or radiotherapy, facilitating the induction of cell death in cancer cells [65,91,92,93,94,95,96,97,98,99]. There are reports suggesting that SMs have the ability to modulate the immune response by increasing the expression of pro-inflammatory cytokines (such as TNF, IL8, IL10, CCL2) and enhancing the immune response [94,95,96]. Another advantage of mimetics is the possibility of non-invasive administration in oral formulations or intravenous infusions [65,91,92,93,94,95,96,97,98]. Despite the ease of drug administration to patients, SM-based therapies distribute well throughout the body, successfully reaching tumor tissues, with accumulation in tumors observed up to 7 days after the end of therapy [91].

However, it is important to consider the adverse effects associated with the use of these mimetics. Clinicians report that the most common adverse reactions in cancers other than HNSCC include fatigue, dizziness, diarrhea, fever, and rashes [65,92,94,98,99]. Nevertheless, in a study on the use of Xevinapant in patients with HNSCC, minimal differences in side effects were noted between the group receiving combination therapy with cisplatin and the group receiving cisplatin and placebo [100,101,102]. The underlying causes of adverse effects associated with the use of Smac mimetics require further investigation and consideration, particularly in therapies exhibiting similar side effects [65,91,92,93,94,95,96,97,98,99,100,101,102].

## 8. Conclusions

This publication presents a comprehensive evaluation of the therapeutic potential of inhibitors of anti-apoptotic proteins (IAPs) in the context of head and neck squamous cell carcinoma (HNSCC). The available studies, including molecular analyses, in vitro experiments, in vivo studies, and clinical trials, were collected and described to determine the impact of IAP inhibitors on apoptotic processes in HNSCC cancer cells. The research demonstrated that IAP inhibitors can effectively overcome resistance to traditional therapies, such as radiotherapy and chemotherapy, by increasing the sensitivity of cancer cells to apoptosis induction. The clinical prospects of using these inhibitors in the treatment of advanced HNSCC, as well as the potential benefits of integrating these compounds with existing therapeutic regimens, were also discussed. Consequently, this study suggests that IAP inhibitors may represent a promising new class of drugs in the fight against HNSCC, particularly in difficult-to-treat cases. However, further research is necessary to fully understand the potential side effects and to determine the optimal conditions for the use of these inhibitors in clinical practice.

## Figures and Tables

**Figure 1 pharmaceuticals-17-01308-f001:**
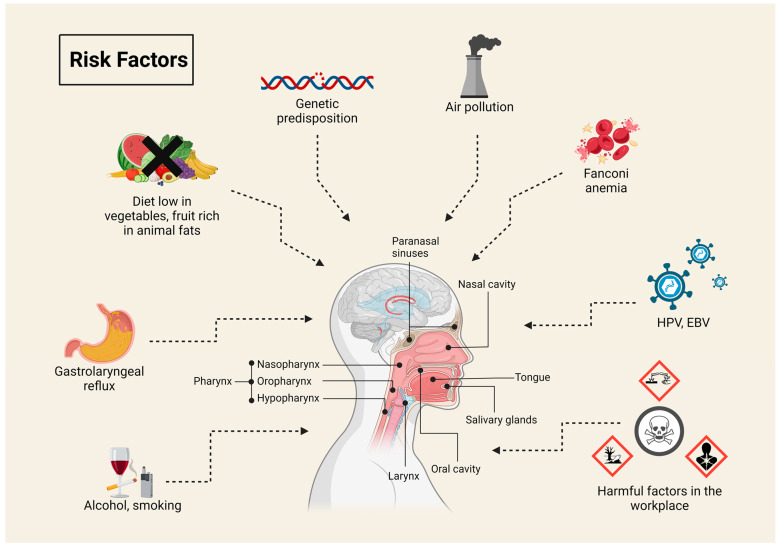
Risk factors of head and neck squamous cell carcinoma. The figure presents the main risk factors for the development of head and neck cancers, such as smoking, alcohol consumption, and viral infections (HPV, EBV). Other significant factors include genetic predisposition, exposure to air pollution, Fanconi anemia, and harmful substances in the workplace. Additionally, a diet low in vegetables and high in animal fats, as well as gastroesophageal reflux, may increase the risk of cancer development.

**Figure 2 pharmaceuticals-17-01308-f002:**
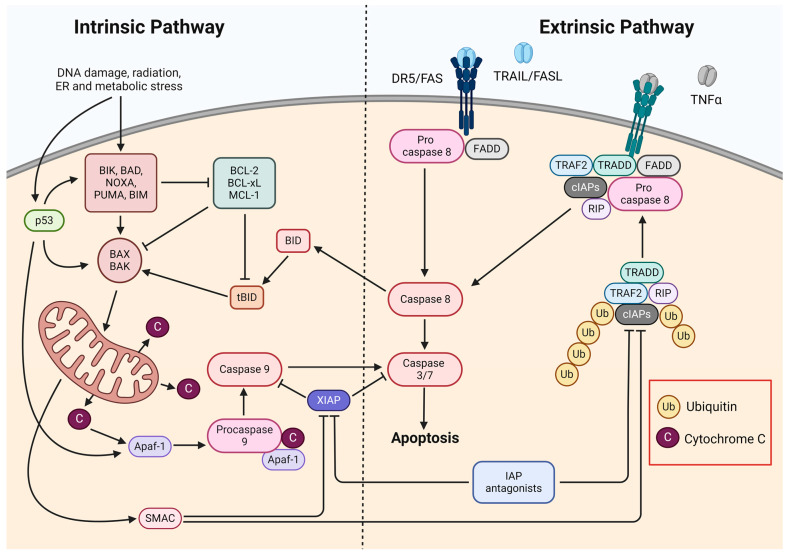
Intrinsic and extrinsic pathway of apoptosis. The figure depicts the intrinsic and extrinsic apoptotic pathways. The intrinsic pathway is triggered by factors like DNA damage or oxidative stress, leading to mitochondrial cytochrome c release, which activates caspase-9 and, subsequently, effector caspases (caspase-3/-7), driving apoptosis. Pro- and anti-apoptotic BCL-2 family proteins regulate this process. The extrinsic pathway is initiated by death ligands (e.g., TNFα, FasL) binding to cell surface receptors, activating caspase-8, which directly activates effector caspases. Caspase-8 also cleaves BID, linking both pathways. Both pathways converge on caspase-3/-7, resulting in apoptosis.

**Figure 3 pharmaceuticals-17-01308-f003:**
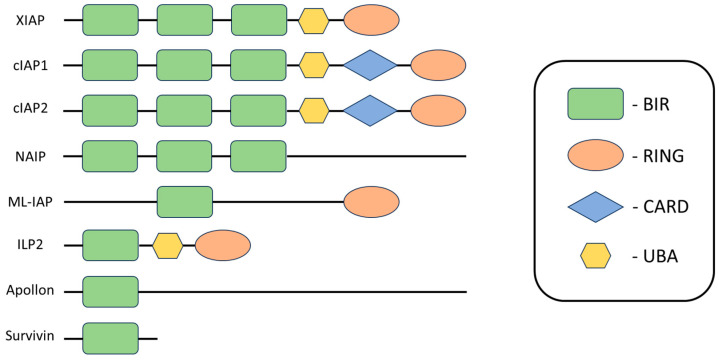
Inhibitor of apoptosis protein family—schematic representation of the structure. BIR—baculovirus IAP repeat; RING—really interesting new gene; CARD—caspase recruitment domain; UBA—ubiquitin conjugating domain [45,47].

**Figure 4 pharmaceuticals-17-01308-f004:**
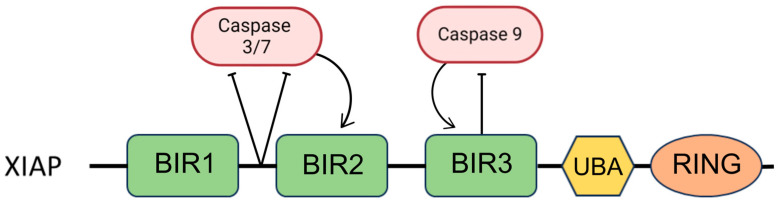
XIAP—sites of interaction and action with caspases. XIAP directly inhibits the activity of effector caspases, such as caspase-3 and -7, by binding its BIR1-BIR2 linker region to their active sites, effectively blocking their function. This interaction is further stabilized by the BIR2 domain, which binds to the exposed IAP-binding motif on these caspases. Additionally, the BIR3 domain of XIAP binds to caspase-9, inhibiting its activation and preventing it from promoting apoptosis [49].

**Figure 5 pharmaceuticals-17-01308-f005:**
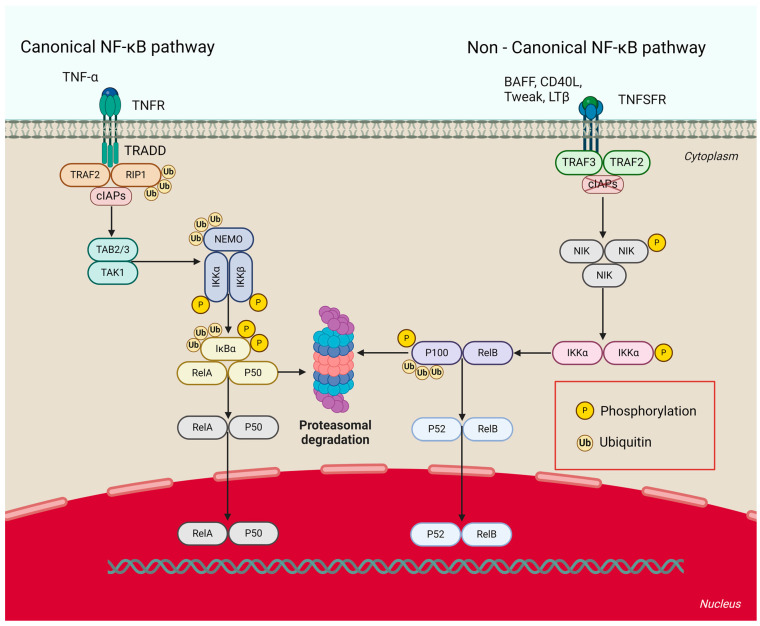
Canonical and non-canonical NF-κB signaling. The figure illustrates the canonical and non-canonical NF-κB pathways. In the canonical pathway, TNF-α binds to TNFR, leading to the recruitment of cIAPs, TRAF2, and RIP1. This complex activates TAK1, which phosphorylates IKKα and IKKβ, promoting the degradation of IκBα and the translocation of the RelA/p50 complex to the nucleus for gene transcription. In the non-canonical pathway, ligands like BAFF and CD40L bind to TNFSFR, resulting in TRAF2 and TRAF3 complex formation. cIAPs facilitate NIK ubiquitination. When cIAPs are inhibited, NIK accumulates, leading to IKKα phosphorylation and the formation of the RelB/p52 complex, which translocates to the nucleus to regulate gene transcription.

**Figure 6 pharmaceuticals-17-01308-f006:**
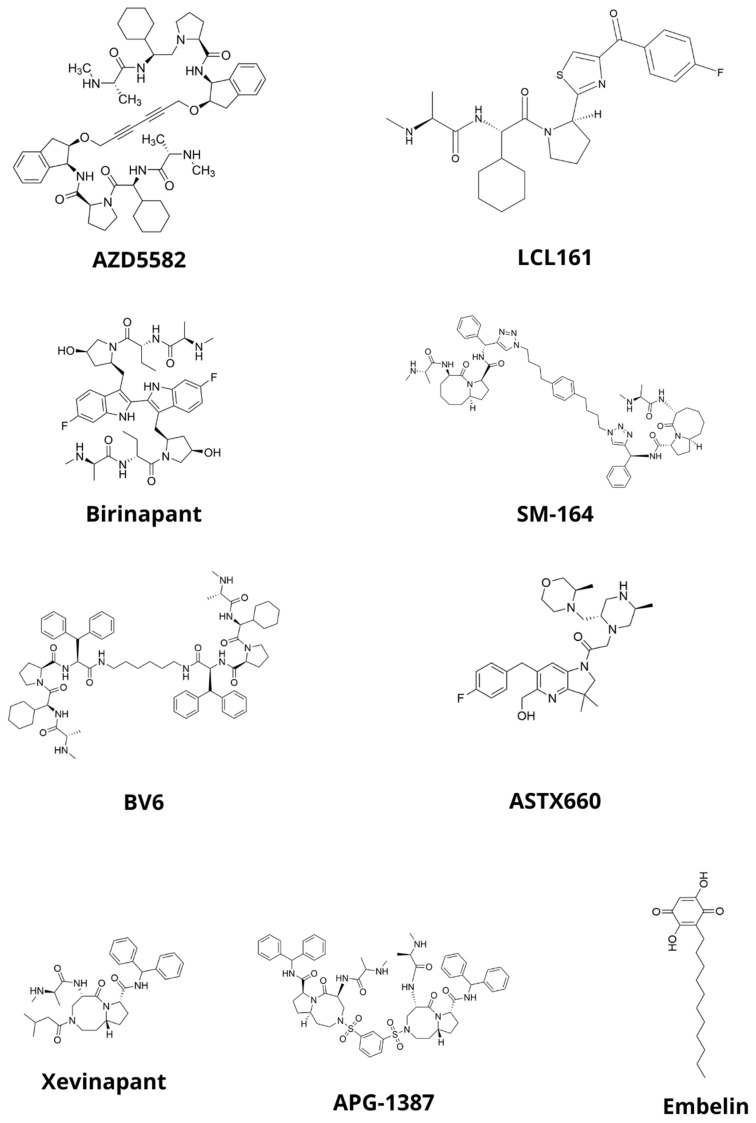
Chemical structure of selected IAP inhibitors.

**Figure 7 pharmaceuticals-17-01308-f007:**
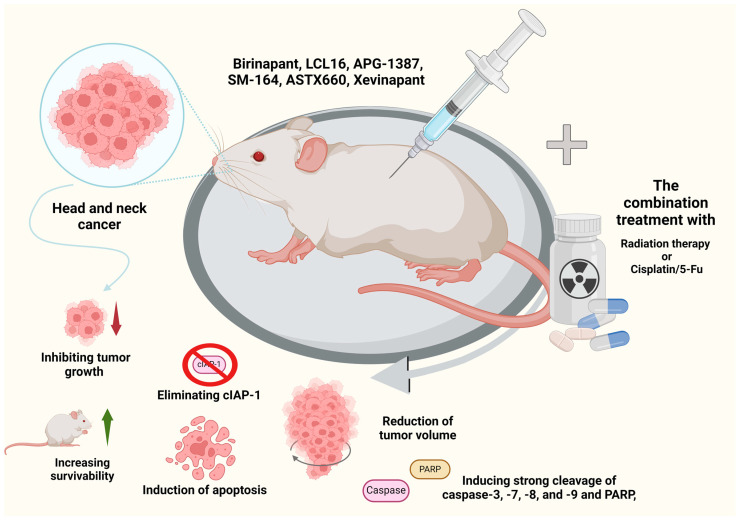
Effects of IAP inhibitors in animal studies. The figure presents a summary of preclinical studies where various Smac mimetics, such as Birinapant, LCL161, APG-1387, SM-164, ASTX660, and Xevinapant, were tested in animal models of head and neck cancer. The treatment involved monotherapy with IAP inhibitors or a combination with standard therapies, including radiation therapy or chemotherapy (e.g., Cisplatin/5-Fu). The main observed treatment effects include tumor growth inhibition, tumor volume reduction, increased survivability, elimination of cIAP-1, induction of apoptosis, caspase activation and PARP cleavage.

**Table 2 pharmaceuticals-17-01308-t002:** Table presents clinical trials focused on the application of IAP inhibitors in patients with head and neck cancer.

IAP	Phase	The Aim of the Clinical Trial	Intervention	Condition	Status	Number
Birinapant	I	The side effects and optimal dose of birinapant administered alongside intensity-modulated re-irradiation therapy in the treatment of patients with HNSCC, who have experienced recurrence at or near the site of the original tumor (locally recurrent), have been under investigation.	+IMRT	Locally Recurrent Head and Neck Squamous Cell Carcinoma Nasopharyngeal Squamous Cell Carcinoma Sinonasal Squamous Cell Carcinoma	Terminated	NCT03803774
Xevinapant	I	Determination of the optimal safe dose of xevinapant administered in combination with radiotherapy and chemotherapy.	+RT +carboplatin or packlitaxel	Head and Neck Cancer Head and Neck Neoplasm Squamous Cell Carcinoma of Head and Neck	Active, not recruiting	NCT06110195
	III	Evaluation of the efficacy and safety of Xevinapant combined with radiotherapy compared to placebo and radiotherapy in participants with resected HNSCC at high risk of recurrence who are ineligible for high-dose cisplatin treatment.	+IMRT	Head and Neck Cancer	Active, not recruiting	NCT05386550
	II	Assessment of the efficacy and safety of Xevinapant in combination with radiotherapy in elderly patients with locally advanced squamous cell carcinoma of the head and neck (LA-HNSCC) involving the oral cavity, oropharynx, hypopharynx, or larynx.	+IMRT	Locally Advanced Head and Neck Squamous Cell Carcinoma	Recruiting	NCT05724602
	Ib	Evaluation of the tolerability and safety of Xevinapant in combination with weekly cisplatin-based chemoradiotherapy for the treatment of patients with unresectable LA-HNSCC eligible for definitive chemoradiotherapy.	+cisplatin +IMRT	Head and neck cancer	Active, not recruiting	NCT06056310
	II	Assessment of the efficacy of adding Xevinapant to standard postoperative chemoradiotherapy in high-risk head and neck cancer patients.	+RT +cisplatin	Head and neck cancer	Active, not recruiting	NCT06145412
	III	Demonstration of the efficacy of Xevinapant in combination with radiotherapy and cetuximab compared to radiotherapy, cetuximab, and placebo in previously untreated participants with LA-HNSCC who are ineligible for high-dose cisplatin therapy.	+IMRT +cetuximab	Squamous Cell Carcinoma of the Head and Neck	Suspended	NCT05930938
	III	Demonstration of the efficacy of Xevinapant (Debio 1143) compared to placebo in combination with chemoradiotherapy in LA-HNSCC	+IMRT +cisplatin	Squamous Cell Carcinoma of the Head and Neck	Active, not recruiting	NCT04459715
	II	Evaluation of whether adding Xevinapant to postoperative adjuvant treatment with cisplatin and radiotherapy, followed by Xevinapant monotherapy, in patients with surgically resected HNSCC with extranodal extension and/or positive margins, will improve disease-free survival.	+IMRT and IGRT +carboplatin +cisplatin	Head and Neck Squamous Cell Carcinoma Hypopharyngeal Squamous Cell Carcinoma Laryngeal Squamous Cell Carcinoma Oral Cavity Squamous Cell Carcinoma Oropharyngeal Squamous Cell Carcinoma Stage III Cutaneous Squamous Cell Carcinoma of the Head and Neck AJCC v8 Stage III Hypopharyngeal Carcinoma AJCC v8 Stage III Laryngeal Cancer AJCC v8 Stage III Lip and Oral Cavity Cancer AJCC v8 Stage III Oropharyngeal (p16-Negative) Carcinoma AJCC v8 Stage IV Cutaneous Squamous Cell Carcinoma of the Head and Neck AJCC v8 Stage IV Hypopharyngeal Carcinoma AJCC v8 Stage IV Laryngeal Cancer AJCC v8 Stage IV Lip and Oral Cavity Cancer AJCC v8 Stage IV Oropharyngeal (p16-Negative) Carcinoma AJCC v8	Withdrawn	NCT06084845
	I/II	A phase I/II randomized study to determine the maximum tolerated dose, safety, pharmacokinetics and antitumor activity of Debio 1143 combined with concurrent chemo-radiation therapy in patients with locally advanced squamous cell carcinoma of the head and neck.	+RT +cisplatin	Advanced Squamous Cell Carcinoma of the Head and Neck	Completed	NCT02022098
Tolinapant	I	Demonstration of the safety and side effects of tolinapant administered in combination with radiotherapy in the treatment of patients with head and neck cancer, who are treatment-naïve, have disease that has spread to nearby tissues or lymph nodes, and are ineligible for cisplatin therapy.	+RT	Head and Neck Carcinoma of Unknown Primary Locally Advanced Head and Neck Squamous Cell Carcinoma Locally Advanced Hypopharyngeal Squamous Cell Carcinoma Locally Advanced Laryngeal Squamous Cell Carcinoma Locally Advanced Nasopharyngeal Squamous Cell Carcinoma Locally Advanced Oropharyngeal Squamous Cell Carcinoma	Recruiting	NCT05245682

IMRT—intensity-modulated re-irradiation therapy; RT—radiotherapy; IGRT—image-guided radiation therapy.

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
