# Peer review of "Multifaceted Evaluation of Inhibitors of Anti-Apoptotic Proteins in Head and Neck Cancer: Insights from In Vitro, In Vivo, and Clinical Studies (Review)"

_pharmaceuticals, 2024, doi:10.3390/ph17101308_

Round 1

Reviewer 1 Report

Comments and Suggestions for Authors

A review manuscript entitile "Multifaceted Evaluation of IAP Inhibition in Head and Neck 2 Cancer: Insights from In Vitro, In Vivo, and Clinical Studies 3" brzykawski is well written manuscript. I have raised following points that may help to strenghten the manuscript further:

1.  In all Figures, there are only titles. Authors may consider to provide concise legends in each figures.

2. Figure 2, authors have missed to provide a concise role of p53 tumor supressor in apoptosis.

3. If possible, authors may provide and illustrate in an additional schematic diagram depicting the interaction sites, and actions of IAP with caspases?

Author Response

„Multifaceted Evaluation of IAP Inhibition in Head and Neck Cancer: Insights from In Vitro, In Vivo, and Clinical Studies (review)”

Kamil Krzykawski, Robert Kubina, Dominika Wendlocha, Robert Sarna, Aleksandra Mielczarek-Palacz

Manuscript Status: Pending minor revisions

The authors greatly appreciate valuable and positive comments provided by Reviewers which enhanced our manuscript. We have submitted a revised version of our manuscript „Multifaceted Evaluation of IAP Inhibition in Head and Neck Cancer: Insights from In Vitro, In Vivo, and Clinical Studies (review)”. In order to prepare a corrected version, we have followed the comments made by the reviewers. We hope that our revisions will be found satisfactory and the manuscript will be suitable for publication.  Please find our answers and acknowledgements to the Reviewers attached below.

Point-by-point answers to the Reviewer 1:

  1. In all Figures, there are only titles. Authors may consider to provide concise legends in each figures.

Reply: Thank you for your valuable suggestion. In accordance with your recommendation, appropriate legends have been added to each figure to provide more detailed explanations of the graphical elements presented. I hope these changes will improve the clarity of the paper and facilitate the understanding of the figures and the data they depict.

  1. Figure 2, authors have missed to provide a concise role of p53 tumor supressor in apoptosis.

Reply: Thank you for your insightful comment. We have slightly modified Figure 2 to better illustrate the role of the p53 tumor suppressor in apoptosis. Our intention was to maintain a concise and focused presentation of the figure while highlighting the essential aspects of p53’s function. We hope the adjustments made enhance the clarity without compromising the compact nature of the graphical representation.

  1. If possible, authors may provide and illustrate in an additional schematic diagram depicting the interaction sites, and actions of IAP with caspases?

    Reply: In accordance with your suggestion, we have included a figure in the manuscript illustrating the binding of caspases by XIAP – the only human IAP protein that exhibits strong affinity for caspases. This figure highlights the key interaction sites of XIAP in inhibiting caspase activity, which should significantly enhance the visualization of the discussed topic.

Reviewer 2 Report

Comments and Suggestions for Authors

Dear Authors,

Thank you very much for the submission of the manuscript entitled "Multifaceted Evaluation of IAP Inhibition in Head and Neck Cancer: Insights from In Vitro, In Vivo, and Clinical Studies (review) ".

This review article seems to be comprehensive, well-structured, and very informative. However, several issues could be improved. 

Please, find my comments and suggestions below.

1. P. 1, line 33

Are electronic cigarettes or vapes more dangerous and oncogenic than traditional tobacco products? Please, indicate it in this section.

2. P. 2, line 46

The Authors highlighted that Epstein-Barr virus infection is a risk factor for HNSCC. Please, compare the risk level of Epstein-Barr virus with Human cytomegalovirus and Herpes simplex virus.

3. Section 1 "Head and neck cancer"

I kindly recommend the Authors adding the novel examples of quantum dots for imaging (diagnosis) and photodynamic therapy for cancer treatment.

4. P. 8, Section 5

The chemical structures of the pharmaceutical agents should be added.

5. Section 5 and Section 6

I kindly recommend the Authors to include several most illustrative figures from the cited literature.

6. P. 19, Section 6

Please, add the summarized table in this section.

In summary, I suggest accepting this manuscript after minor revisions.

Author Response

„Multifaceted Evaluation of IAP Inhibition in Head and Neck Cancer: Insights from In Vitro, In Vivo, and Clinical Studies (review)”

Kamil Krzykawski, Robert Kubina, Dominika Wendlocha, Robert Sarna, Aleksandra Mielczarek-Palacz

Manuscript Status: Pending minor revisions

The authors greatly appreciate valuable and positive comments provided by Reviewers which enhanced our manuscript. We have submitted a revised version of our manuscript „Multifaceted Evaluation of IAP Inhibition in Head and Neck Cancer: Insights from In Vitro, In Vivo, and Clinical Studies (review)”. In order to prepare a corrected version, we have followed the comments made by the reviewers. We hope that our revisions will be found satisfactory and the manuscript will be suitable for publication.  Please find our answers and acknowledgements to the Reviewers attached below.

Point-by-point answers to the Reviewer 2:

This review article seems to be comprehensive, well-structured, and very informative. However, several issues could be improved.

  1. 1, line 33

Are electronic cigarettes or vapes more dangerous and oncogenic than traditional tobacco products? Please, indicate it in this section.

Reply: Thank you for your suggestion. We have added a section to the manuscript on page 1, addressing electronic cigarettes and vaporizers, highlighting these devices and the existing reports regarding their potential risks.

  1. 2, line 46

The Authors highlighted that Epstein-Barr virus infection is a risk factor for HNSCC. Please, compare the risk level of Epstein-Barr virus with Human cytomegalovirus and Herpes simplex virus.

Reply: Thank you for your valuable comment. On page 2, we have added a section to the manuscript describing the impact of the EBV virus, as well as the other mentioned viruses, such as Human Cytomegalovirus and Herpes Simplex Virus, in the context of the risk of developing HNSCC.

  1. Section 1 "Head and neck cancer"

I kindly recommend the Authors adding the novel examples of quantum dots for imaging (diagnosis) and photodynamic therapy for cancer treatment.

Reply: We appreciate your insightful recommendation. On page 3, line 105, we have added content highlighting the promising properties of quantum dots in diagnostics, and on page 4, line 137, we discuss their use in photodynamic therapy for cancer treatment.

  1. 8, Section 5

The chemical structures of the pharmaceutical agents should be added.

Reply: Thank you for your valuable comment. We are pleased to inform you that a figure illustrating the structures of the discussed compounds has been added on page 11.

  1. Section 5 and Section 6

I kindly recommend the Authors to include several most illustrative figures from the cited literature.

Reply: Thank you very much for your suggestion regarding the inclusion of illustrative figures from the cited literature. We truly appreciate your input. However, we aimed to avoid using figures from other authors due to copyright considerations. 

  1. 19, Section 6

Please, add the summarized table in this section.

Reply: Thank you very much for your suggestion regarding the addition of a summarized table in this section. We appreciate your attention; however, we would prefer not to include such a table, as we are concerned it might be less accessible for readers.

At the end of this section, there is a figure that provides a general summary of the in vivo study results, which we believe may be clearer and more understandable. We hope this format of data presentation will be sufficient.

Reviewer 3 Report

Comments and Suggestions for Authors

This manuscript by Krzykawski et al. reviews the use of inhibitors of antiapoptotic proteins (IAP) in the context of head and neck cancer. The manuscript is well structured, introducing the disease, the apoptosis process and the role of IAP and the possible utility of their inhibition as a therapeutic strategy.

I believe that the manuscript deserves to be published in Pharmaceuticals as it provides comprehensive and up-to-date information on the topic for interested readers.

However, I would just like to make some comments to the authors that could help improve the article in some way:

IAP inhibitors have been introduced into clinical trials against different tumor types other than head and neck cancer. Briefly, the authors could summarize the main findings of these studies to provide context for applicability in head and neck cancer. It would also be interesting to provide data on the expected side effects, as they have been observed in these clinical trials.

Although the manuscript cites several clinical trials that are at different stages of completion, I miss information and conclusions obtained from the clinical trial NCT02022098, which has already been completed and for which there are several publications (PMID: 32758455; PMID: 36796234; PMID: 37439181) and other related ones (PMID: 37777832).

Finally, all clinical trials reviewed concern IAP inhibitors plus radiotherapy or conventional chemotherapy. Are the authors aware of any planned clinical trials in head and neck cancer to test the combination of IAP inhibitors with immune checkpoint inhibitors? Preclinical studies in multiple myeloma suggest increased efficacy (PMID: 27841872) and results of a clinical trial in colorectal and pancreatic cancer have recently been published (PMID: 38502104).

Author Response

„Multifaceted Evaluation of IAP Inhibition in Head and Neck Cancer: Insights from In Vitro, In Vivo, and Clinical Studies (review)”

Kamil Krzykawski, Robert Kubina, Dominika Wendlocha, Robert Sarna, Aleksandra Mielczarek-Palacz

Manuscript Status: Pending minor revisions

The authors greatly appreciate valuable and positive comments provided by Reviewers which enhanced our manuscript. We have submitted a revised version of our manuscript „Multifaceted Evaluation of IAP Inhibition in Head and Neck Cancer: Insights from In Vitro, In Vivo, and Clinical Studies (review)”. In order to prepare a corrected version, we have followed the comments made by the reviewers. We hope that our revisions will be found satisfactory and the manuscript will be suitable for publication.  Please find our answers and acknowledgements to the Reviewers attached below.

Point-by-point answers to the Reviewer 3:

IAP inhibitors have been introduced into clinical trials against different tumor types other than head and neck cancer. Briefly, the authors could summarize the main findings of these studies to provide context for applicability in head and neck cancer. It would also be interesting to provide data on the expected side effects, as they have been observed in these clinical trials.

Reply: Thank you for your suggestion. In accordance with your recommendation, we have added a section summarizing the main findings from clinical trials involving IAP inhibitors in other tumor types. Additionally, we have included data on the expected side effects as observed in these trials. The updated section can be found on page 27, line 1019. We hope this addition enhances the clarity and relevance of the manuscript.

Although the manuscript cites several clinical trials that are at different stages of completion, I miss information and conclusions obtained from the clinical trial NCT02022098, which has already been completed and for which there are several publications (PMID: 32758455; PMID: 36796234; PMID: 37439181) and other related ones (PMID: 37777832).

Reply: Thank you for your valuable feedback. We have added a section describing the results of the NCT02022098 clinical trial, as you suggested. This new information can be found on page 27, line 1003. We hope this addition addresses your comments and enhances the overall quality of the manuscript.

Finally, all clinical trials reviewed concern IAP inhibitors plus radiotherapy or conventional chemotherapy. Are the authors aware of any planned clinical trials in head and neck cancer to test the combination of IAP inhibitors with immune checkpoint inhibitors? Preclinical studies in multiple myeloma suggest increased efficacy (PMID: 27841872) and results of a clinical trial in colorectal and pancreatic cancer have recently been published (PMID: 38502104).

Reply: Thank you for your valuable comment. After conducting a thorough search of the Clinical Trials database, we were unable to find any studies specifically investigating the combination of IAP inhibitors with immune checkpoint inhibitors for the treatment of head and neck cancers. We appreciate your reference to preclinical studies in multiple myeloma and recent results in colorectal and pancreatic cancer, which highlight the potential of this therapeutic combination.